# Prefrontal control of superior colliculus modulates innate escape behavior following adversity

Ami Ritter [1,2], Shlomi Habusha[1,2], Lior Givon [1,2], Shahaf Edut[1,2] & Oded Klavir [1,2] ✉

Innate defensive responses, though primarily instinctive, must also be highly adaptive to changes in risk assessment. However, adaptive changes can become maladaptive, following severe stress, as seen in posttraumatic stress disorder (PTSD). In a series of experiments, we observed long-term changes in innate escape behavior of male mice towards a previously non-threatening stimulus following an adverse shock experience manifested as a shift in the threshold of threat response. By recording neural activity in the superior colliculus (SC) while phototagging specific responses to afferents, we established the crucial influence of input arriving at the SC from the medial prefrontal cortex (mPFC), both directly and indirectly, on escape-related activity after adverse shock experience. Inactivating these specific projections during the shock effectively abolished the observed changes. Conversely, optogenetically activating them during encounters controlled escape responses. This establishes the necessity and sufficiency of those specific mPFC inputs into the SC for adverse experience related changes in innate escape behavior.

Innate defensive responses are crucial for survival, allowing split-second reactions to potential threats[1,2]. While predominantly instinctual, threat responses exhibit flexibility and adapt to risk assessment by the organism, based on factors such as the perceived threat value of an agent in the environment, environmental conditions, internal states and prior knowledge of the organism[3,4]. Modifying threat response is generally an adaptive process[5–7], preventing unnecessary costs from the organism (in energy spent, feeding opportunities lost etc.)[8]. However, threat response flexibility could become maladaptive after severe stress. Posttraumatic stress disorder (PTSD) is associated with long lasting changes after an exposure to a traumatic event, including changes in response to sensory information expressed in enhanced threat detection and reaction, and in reduced arousal thresholds and filter mechanisms[9–12]. As in human patients, major, long-lasting behavioral changes are evident in animals after a single exposure to a severe stressor[13–16].

Escape behavior is a classical innate defensive action which, although required to be fast and almost reflex-like, includes intricate processing and computations to minimize reaction time while maximizing success or adaptivity by considering as much information as possible[3]. As such, in many species it is carried out by subcortical structures, directly receiving sensory information (bypassing sensory cortices) and engaging action, which allows for rapid and intense responses[17]. An important neural hub in managing escape is the superior colliculus (SC)[18–20], a subcortical structure which receives direct retinal input on its superficial layers and generates premotor activity in the deeper layers[21,22], making it a subcortical sensorimotor integration center. Integration in the SC was shown to be crucial for linking sensory threat information and escape[23]. Importantly, while adaptation of escape after learning has also been suggested to occur in SC neurons[5,18], the source of this neural adaptation remains unknown. As escape response is carried out through rapid, semi-instinctive

[1]School of Psychological Sciences, The University of Haifa, Haifa, Israel. [2]The Integrated Brain and Behavior Research Center (IBBRC), University of Haifa, Haifa, Israel. ✉e-mail: klavirod@gmail.com

systems, such long lasting modification may involve changes in higher control systems regulating threat responses.

The medial prefrontal cortex (mPFC) is a complex and heterogeneous region. It is mostly accepted that the rodents subregions of the mPFC – the infralimbic (IL), prelimbic (PL) and cingulate cortex, are in many aspects the homologues of Brodmann's areas 32, 25 and the anterior parts of area 24 in the primate[24–26]. The mPFC integrates previously learned emotional values into decisions and exerting adaptive control over different actions and behaviors[27–29] depending on the specific subcortical circuit receiving mPFC input[30]. For example, activating mPFC projections to the dPAG, a specific relay station in the defensive behavior pathway, increased specific stress related actions, such as digging and marble burying[31]. The SC is also regulated by the mPFC, both directly[32,33] and indirectly through the basal-ganglia[34,35], where it is considered the cortical hub of the associative loop[36], and involved in goal-directed action control[37]. Interestingly, it was recently demonstrated that this two-way cortical control of the SC could be traced to the same neurons in another frontal cortex region – the somatic sensorimotor area, which sends projections both directly to the SC and to the dorsal striatum which affects the SC via the cortico-basal ganglia-pathway[32].

In this work, in order to find the mechanism by which escape threat response is modified after a significant aversive experience, we recorded SC neurons of freely behaving mice confronted with an erratically moving robo-beetle[38]. Behavior was recorded before and at two time points after a significant, unexpected and inescapable foot shock – a procedure known to elicit anxiety related long term changes in behavior[13,15]. Following the aversive experience, we found long term changes in innate escape behavior – an increase in amount and an early onset, corresponding to changes in the threshold of threat tolerance. Photo-tagging of incoming projections into the SC indicated that the changes were induced by mPFC neurons which project to both dorsomedial striatum (DMS) and directly to the SC. Examining general inputs into the SC either through general mPFC neural population or through all basal ganglia channeled input via the substantia nigra pars reticulata (SNr) seemed to dilute the change. Chemogenetically inhibiting mPFC neurons projecting to DMS during the aversive experience prevented the long-term change in escape threshold, while optogenetically activating their SC projections during Beetle Mania Task (BMT) modified the threshold according to the distance-dependent stimulation. Our results point to specific top-down influence through a specialized circuit as necessary and sufficient in mediating enhanced threat detection after a significant aversive experience.

## Results

All descriptive statistics and statistical analyses in full can be found in the supplementary statistical table. All brain schemes for reconstructions are from ref. 39.

### Significant, unexpected, and inescapable foot shock induces long term changes in behavior and specifically in escape behavior

We first set out to test whether a severe, distressful experience (see 'shock phase' in the methods section) can alter the animal's behavioral outcomes in the context of a previously non-threatening stimulus and environment. To this aim we adapted the Beetle Mania Task (BMT)[38] to our needs and changed it into a within subject design (Fig. 1a). This allowed us to examine behavioral changes of the same animal in a semi-ethological environment, where the mouse encounters a smaller erratically moving robo-beetle. This moving, interacting stimulus has no preset threat value, nor does it harm the mouse in any way during interaction. Hence, it does not become a threat by interaction alone (Fig. 1b). The mouse encounters the robo-beetle on three occasions – the first is prior to the shock, and then at

two time points thereafter (Fig. 1a). All interactions were recorded using a camera, set up above the arena, and locations and poses of both, mouse and robo-beetle, were identified and estimated using DeepLabCut2[40,41]. By measuring the vectors of velocity and distance between mouse and beetle, an escape was defined as an abrupt change in the mouse's velocity, increasing the distance between itself and an approaching beetle (Fig. 1c). As a more general assessment for anxiety, we chose to measure the relative time the animal spent in the arena center (central 40% of arena range). To test whether any of the changes could be explained as resulting from the mere encounters with the beetle, we compared the mice that underwent the foot-shock phase to a control group undergoing the same paradigm but without shock administration (placed in the shock boxes for the same amount of time without receiving any shock). A repeated and between-group mixed two-way ANOVA was conducted, considering each behavioral measure in the different measuring days as a repeated measure variable and Control/Shock as a between subject variable. A total of twenty mice in the shock group and thirteen in the control group had data in all measuring time points and could be included in the analysis. We found that exposure to a significant aversive event resulted in increased numbers of escape responses as measured 7 days thereafter; a change which was maintained also 21 days post-shock and appeared only in the shock group (Fig. 1d; a mixed model two-way ANOVA showed a significant Time X group interaction effect; $F(2,60) = 8.5$, $p < 0.001$; post-hoc test revealed a significant difference at the shock group from the pre-shock baseline at the 7th day; $p < 0.01$ and the 21st day $p < 0.001$ and differences between shock group and control on the 7th day and the 21st day from shock, both $p < 0.001$). Not only were there more escape events when measured 7 and 21 days after the shock, but we found that shock mice also maintained a larger safety distance from the beetle as they detected it as a threat earlier and escape onsets occurred at a farther distance from the approaching beetle. (Fig. 1e; a trend of Time X group interaction; $F(2,60) = 2.56$, $p = 0.085$; post-hoc test revealed a significant difference at the shock group from the pre-shock baseline at the 7th day; $p < 0.05$ and the 21st day $p < 0.05$ and differences between shock group and control on the 7th day $p < 0.05$). The changes in escape behavior occurred specifically after the shock administration and were maintained for weeks. Another maintained change was general anxiety as only shocked mice spent significantly less time in the central 40% of the arena on both the 7th and 21st day after the shock, as compared to baseline. (Fig. 1f; a mixed model two-way ANOVA showed a significant main effect of Time ($F(2,60) = 4.14$; $p < 0.05$) and a trend of Time X group interaction; $F(2,60) = 2.51$, $p = 0.085$; post-hoc test revealed a significant difference at the shock group from the pre-shock baseline at the 7th day; $p < 0.0001$ and the 21st day $p < 0.001$ and differences between shock group and control on the 7th day and the 21st day from shock, both $p < 0.05$).

To isolate the effect of enhanced defensive distance, we further divided the escape events of the shock group into escapes, initiated when the approaching beetle was in the line of sight of the mouse, termed – preventive escapes (PE; Fig. 1g) and escapes initiated when beetle was out of sight (usually by bumping into the mouse), termed – reactive escapes (RE; Fig. 1j). We found that the increase in the number of escape events occurred in both types of escape. A repeated measures one-way ANOVA showed main effect of time for both preventive (Fig. 1h; $F(2,30) = 6.651$, $p < 0.05$; post-hoc test 7th and 21st day both $p < 0.01$) and for reactive escapes (Fig. 1k; $F(2,28) = 5.682$, $p < 0.01$; post-hoc test 7th day $p < 0.05$ and 21st day $p < 0.01$). However, changes in escape characteristics depending on visual detection, such as the safety distance, indeed occurred only in preventive escapes (Fig. 1i; $F(2,38) = 3.401$, $p < 0.05$; post-hoc test 7th and 21st day both $p < 0.05$). No such effect was found for reactive escapes. (Fig. 1l; No significant effects of time in a repeated measures ANOVA ns).

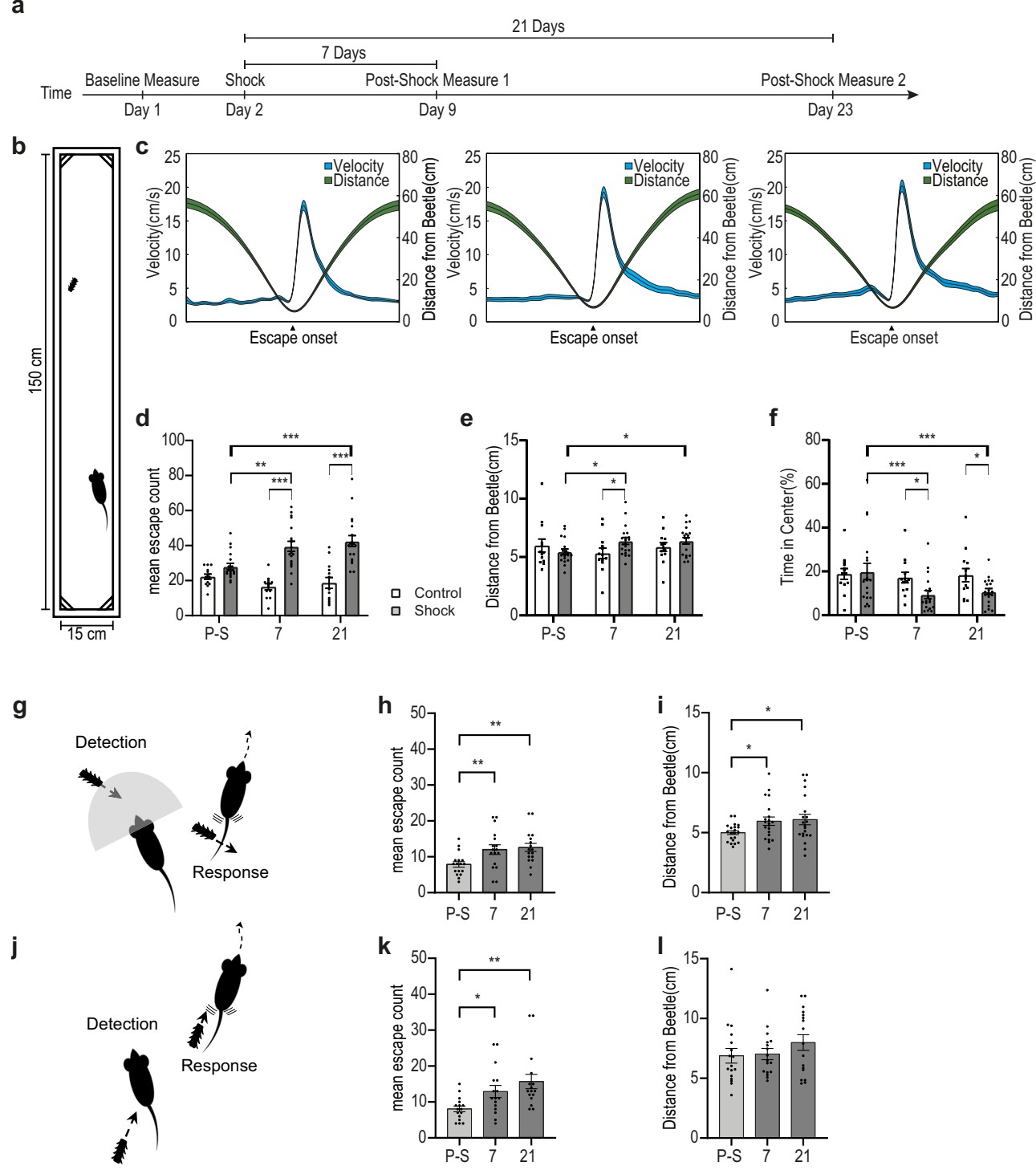

## Superior colliculus (SC) neurons involved in escape initiation, maintain a longer safety distance long after experience with an inescapable foot shock

Searching for the neural changes underlying the early escape response within an enhanced threat detection distance, we implanted a custom made optrode (see methods section) in the SC of mice running the task (Fig. 2a, top). We then recorded a total of 1395 units from the SC of 31 mice (see Fig. 2a, bottom for the reconstruction of fiber locations). 80% (1115) of the recorded units significantly increased firing rate around events of any abrupt accelerations by the mouse and were termed "responsive to acceleration", out of which 77% (864 units) were

responsive specifically during preventive escapes, and 54% (630 units) responded during reactive escapes (Fig. 2b). A vast majority (87%) of acceleration-responsive units initiated neural response prior to the acceleration itself (Fig. 2c, d), indicating an involvement of those units in escape initiation as was previously suggested[42]. This distribution was solidly maintained when looking at the responses of recorded SC units at all three time-points of the task and even shifted to further precede acceleration on the 21st day after the shock. (Fig. 2e left; A Kruskal–Wallis test for assessing the differences in the distributions of neural responses initiation times relative to escape onset revealed a significant main effect of time H(2) = 49.65, $p < 0.001$; post-hoc test

**Fig. 1 | Exposure to severe adversity results in lasting changes in behavior.**
**a** Behavioral paradigm and time schedule. **b** Schematic depiction of the testing arena with the animal (bottom) and the beetle (top) to scale. **c** Peri-event time histograms depicting the animal's velocity in cm/s (blue, left Y-axis) and the distance between mouse and beetle in cm (green, right Y-axis) around escape onsets, assessed pre-shock, 7 days and 21 days following footshock, left to right respectively. **d–f** Comparisons of behavioral parameters at three time points in shocked ($n = 19$) and non-shocked ($n = 13$) animals showing long-term behavioral changes after the shock day, specifically in shocked mice. All mixed Two Way ANOVA with Tukey's post-hoc comparisons. **d** Increase in the number of escapes from beetle in shocked group only (P-S vs. 7 − $p = 0.007$, P-S vs. 21 − $p < 0.001$). **e** Increase in the distance to beetle at escape onset in shocked group only (P-S vs. 7 − $p = 0.047$, P-S vs. 21 − $p = 0.043$). **f** Decrease in the time spent in the arena center in shocked group only (P-S vs. 7 − $p = 0.007$, P-S vs. 21 − $p = 0.025$). **g** Depiction of our definition for preventive escapes. The sketch displays a mouse and an approaching beetle. The

gray half circle indicates the animal's view orientation and angle. An escape response following visual detection of the beetle by the animal was termed preventive escape (PE). **h** Increase in the number of PE's. Repeated One-Way ANOVA with Tukey's ($n = 16$ P-S vs. 7 − $p = 0.017$, P-S vs. 21 − $p = 0.006$). **i** A greater distance from the beetle at PE onset following footshock ($n = 20$ P-S vs. 7 − $p = 0.047$, P-S vs. 21 − $p = 0.04$). **j** An illustration depicting our definition of reactive escape (RE) − the contrasting subtype for PE. The sketch displays a mouse with a beetle approaching from its rear, prompting the animal to escape once detected. **k** An increase in the number of post-shock RE's persisting over time. Repeated One-Way ANOVA with Tukey's ($n = 15$ P-S vs. 7 − $p = 0.039$, P-S vs. 21 − $p = 0.005$). **l** No change in distance at RE onset was observed ($n = 16 − p = 0.173$). In each plot, bars represent means of the corresponding measures with error bars showing ±SEM. In peri-event histograms, each line represents the overall average of the corresponding measure with shaded ± SEM. Asterisks indicate significant post hoc comparisons (*$p < .05$; **$p < .01$; ***$p < .001$). Source data are provided as a Source Data file.

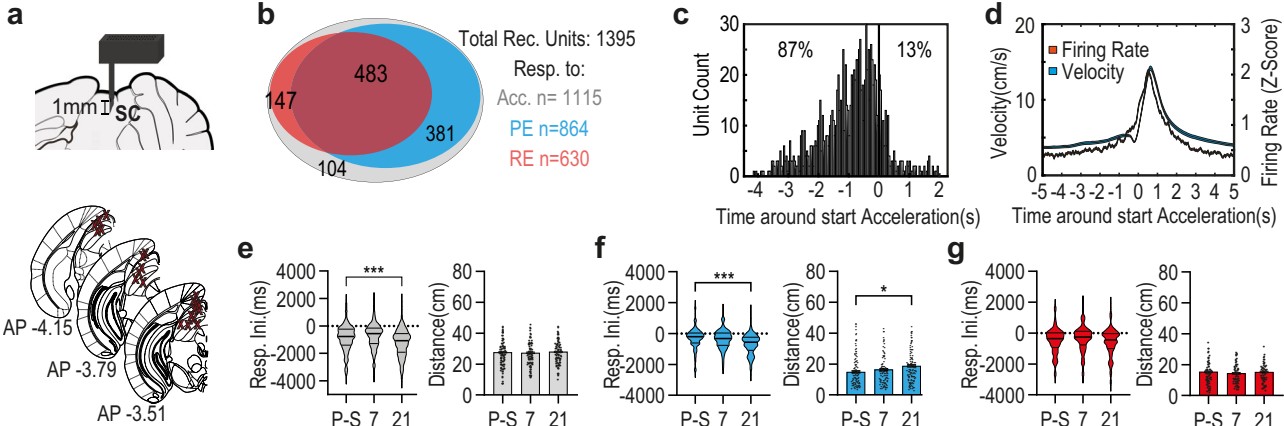

**Fig. 2 | Altered safety distance following footshock is mildly expressed in the neural activity of the general population of escape related SC neurons. a** Top: Schematic depiction of the optrode design and placement unilaterally in the SC. Bottom: Anatomical target schemes for all animals included in the experiment. X's mark histologically confirmed most ventral position of the implant, recording electrodes spread in a brush-like form up to 1 mm below the X. **b** Diagram representing the distribution of the number of acceleration responsive units with respect to whether they respond to PE, RE, both or neither. **c** Histogram of neural response initiation time (s) in relation to acceleration onset (at time 0) for all acceleration-responsive units recorded in the SC. **d** Averaged neural response − red (z-score of firing rate, right Y-axis) and averaged velocity − blue (cm/s, left Y-axis) around acceleration onset (at time 0, X-axis). Lines represent the overall average of the corresponding measure with shaded ±SEM. **e** Left: In pooled acceleration-responsive units, neural response initiation times (ms) relative to acceleration onset undergoes a shift afore in distribution 21 days following footshock. Kruskal-Wallis with Dunn's ($n = 443$ pre-shock; $n = 362$ on 7th-day; $n = 310$ on the 21st-day; Pre-shock vs. 7th-day $p = 0.067$; Pre-shock vs. 21st-day $p < 0.001$). Right: No change

in distance (cm) from the beetle at response initiation time is observed. One-way ANOVA ($n = 370$ pre-shock; $n = 279$ on 7th-day; $n = 283$ on the 21st-day, $p = 0.984$). **f** PE responsive units − left: Spike initiation times (ms) relative to escape onset undergoes a shift afore in distribution. Kruskal-Wallis with Dunn's ($n = 333$ pre-shock; $n = 272$ on 7th-day; $n = 259$ on the 21st-day; Pre-shock vs. 7th-day $p = 0.067$; Pre-shock vs. 21st-day $p < 0.001$). Right: An increase in distance (cm) from beetle corresponding to response initiation time 21 days following footshock. One-Way ANOVA with dunnet's ($n = 237$ pre-shock; $n = 203$ on 7th-day; $n = 220$ on the 21st-day, P-S vs. 21−$p = 0.016$). **g** RE responsive units − left: no change in Spike initiation times (ms) relative to escape onset. Kruskal-Wallis ($n = 250$ pre-shock; $n = 202$ on 7th-day; $n = 177$ on the 21st-day; $p = 0.107$). Right: no change in distance at neural response initiation time. One-way ANOVA ($n = 174$ pre-shock; $n = 123$ on 7th-day; $n = 128$ on the 21st-day, $p = 0.260$). In each plot, bars represent means with error bars showing ±SEM. In violin graphs, the bold midlines represent medians with IQR in regular lines above and below them. Asterisks indicate significant comparisons (*$p < .05$; ***$p < .001$). Source data are provided as a Source Data file.

21st day $p < 0.001$). While the distance from the beetle at the time of neural response initiation preceding acceleration in acceleration-responsive units was maintained around the same distance (Fig. 2e right; No significant main effect of time was found $F_{(2,929)} = 0.0162$, ns). To further study if this shift is related to escape behavior, we looked specifically at escapes through the prism of maintaining a defensive distance and separated to preventive and reactive escapes. In preventive escapes we found that the change in safety distance as expressed in the behavioral results (Fig. 1n) is mildly expressed in the neural response as distribution moves further afore acceleration, slightly on the 7th and significantly on the 21st day post shock (Fig. 2f left; Kruskal-Wallis test revealed a significant main effect of time $H(2) = 33.05$, $p < 0.001$; post-hoc test 21st day $p < 0.001$), as well as the distance from the robo-beetle at the time of neural response onset (Fig. 2f right; one way ANOVA revealed a significant main effect

of time $F_{(2,657)} = 4.408$, $p < 0.05$; post-hoc test 21st day $p < 0.05$). This however was not the case in reactive escapes where safety distance remained constant. (Fig. 2g left and right; $H(2) = 4.464$, ns; as well as $F_{(2,422)} = 1.351$, ns). As the response shift of the units appeared in PE and not in RE units, it was essential to test whether it is indeed related to the initiation of an escape response or is it a sensory alert response to a dashing robo-beetle. Responses of PE units to approaching robo-beetle were separated into cases where the approaching beetle was followed by an escape action and to cases were no escape action followed approach. We found that the approaching beetle per se did not evoke a response compared to approach followed by escape as evident by the 40 ms bin PSTH comparing the activity around approach (Supplementary Fig. 1a), and the response during post approach 400 ms response window was significantly higher when escape followed the approaching beetle (as revealed by a within subject ttest

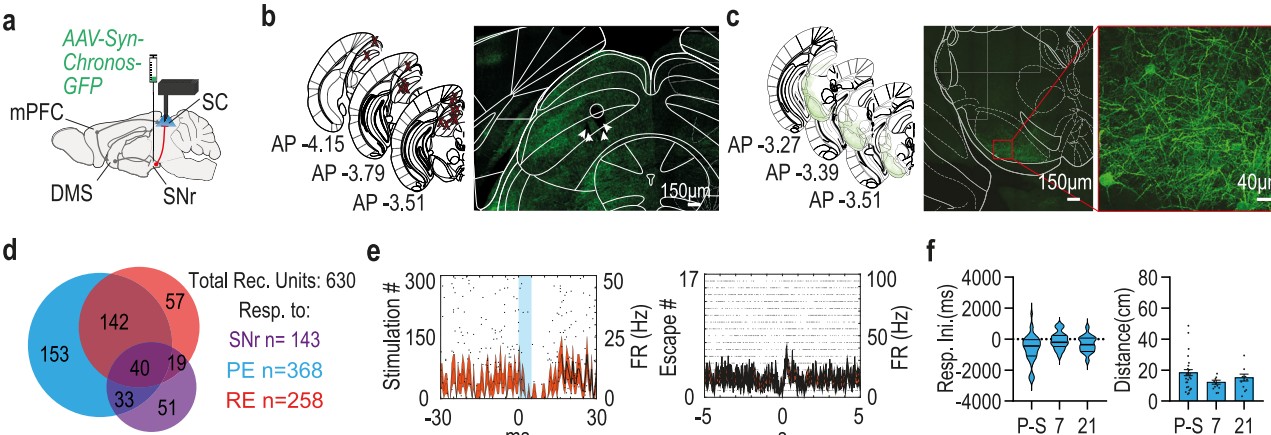

**Fig. 3 | Altered safety distance following footshock is not expressed in the neural activity of specific SNr responsive escape related SC neurons.**
**a** Schematic depiction of optrode location (SC) and viral injection site (SNr). **b** The left illustration depicts anatomical target schemes for all 15 animals included in the experiment (Representative images in this figure present results successfully acquired in all included animals). X's mark histologically confirmed most ventral position of the implant recording electrodes spread in a brush like form up to 1 mm below the X. Right: representative coronal section showing the SC with traces of implant lesion and neural projections from the SNr. White circle marks the suspected bottom of the optic fiber; white arrowheads indicate possible recording electrode location. **c** The left illustration depicts overlaid, histologically confirmed SNr viral expression areas from all included animals. Center: representative coronal section showing viral expression in the SNr. Right: enhanced SNr to see the projecting neurons. **d** Diagram representing the distribution of all recorded units in the experiment with respect to SNr responsiveness, as well as PE and RE related activity.

Numbers on the Venn diagram correspond to unit count. **e** A raster plot overlaid with a PSTH showing a typical response of SNr responsive SC neuron to light stimulation (left) and to PE instances (right), each line represents the overall average of the corresponding measure with shaded ± SEM. **f** PE related SNr responsive SC units – left: Response initiation times almost exclusively precede PE onsets, but the distribution of response initiation times (ms) relative to PE onsets remained unchanged following footshock. Kruskal-Wallis ($n$ = 38 pre-shock; $n$ = 19 on 7th-day; $n$ = 16 on the 21st-day; $p$ = 0.056). Right: No significant change was observed in distance (cm) from the beetle at the time of response initiation time of those neurons. One-way ANOVA ($n$ = 29 pre-shock; $n$ = 13 on 7th-day; $n$ = 11 on the 21st-day, $p$ = 0.144). In violin graphs, the bold midlines represent medians with IQR in regular lines above and below them, bars represent means of corresponding measures with error-bars showing ±SEM. Source data are provided as a Source Data file. Parts of the Illustrations were generated with Biorender.com.

t(863) = 17.72; $p$ < 0.0001; Supplementary Fig. 1b). However, it seems that the neural response does depend on sensory information as well as we also found a response bias with higher responses to threat arriving from the side contralateral to the recorded SC neurons as evident by the 40 ms bin PSTH comparing the activity around escape initiation (Supplementary Fig. 1c), also as the response during escape 400 ms response window was significantly higher when escape followed an approaching beetle from the contralateral side (as revealed by a within subject ttest t(840) = 4.208; $p$ < 0.0001; Supplementary Fig. 1d).

While the persistent correlation between escape onsets and the preceding neural response suggests a direct involvement of SC neurons in the initiation and/or upkeep of escape behavior, the shift in the neural activity following shock, suggests upstream modulation of the SC involvement.

### Post-shock changes in preventive escape related activity does not map onto SC neurons receiving direct SNr Projections
To identify the neural inputs supporting the changes in the response timing of escape-SC neurons following the fearful event, we expressed an opsin in SC afferents while photo-tagging the neural response in the SC. We started by examining the output from the basal ganglia to the SC through the inhibitory projections from the SNr. A viral vector driving the expression of an opsin (pAAV-Syn-Chronos-GFP) was injected to the SNr and the optrode was implanted in the SC of mice running the task (Fig. 3a). Using the optrode we recorded the units from the SC of 15 mice (see Fig. 3b for reconstruction of the location of the fibers – left; and example microscope slice showing optrode damage+ opsin expression – right). SNr expression was also validated (Fig. 3c for reconstruction of the opsin expression in the SNr – left; and example microscope slice – center and right). As shown in Fig. 3d Out of 630 recorded units, 23% (143 units) were found responsive to SNr. 51% (73 units) of SNr responsive units were responsive specifically to PE

(Fig. 3e) and 41% (59 units) were responsive specifically to RE. The distribution of responses initiation was solidly maintained prior to acceleration at all three time-points of the task without a difference between them. (Fig. 3f left; medians of −340, −240 and −340 ms, respectively, with non-significant main effect of Kruskal-Wallis test H(2) = 0.909; ns). No difference was also found in the distance to the beetle at neural response initiation preceding acceleration (Fig. 3f right; No significant main effect of time was found F(2,50) = 0.247, ns). The finding that the vast (73%) majority of SNr responsive SC units initiate response prior to PE suggests involvement of the SNr in escape initiation through its direct connection to the SC. As the temporal and spatial nature of this response stay stable following shock phase, this synapse does not seem to play a unique role in the behavioral and neural activity changes.

### Post-shock changes in preventive escape related activity tracks down to SC neurons receiving direct and indirect input from a specific cohort of dorsomedial striatum (DMS) projecting medial prefrontal cortex (mPFC) neurons
To find out whether it is the prefrontal neural input which induces the changes in the response timing of escape-SC neurons following the fearful event, we tested two pathways by which the same mPFC neurons might exert influence over the SC, the first is through the basal-ganglia via the mPFC-DMS vast projections, the second is directly via mPFC-SC projections. In order to identify these unique inputs, we first injected a retrograde viral vector driving the expression of the Cre into the DMS (AAVrg-Ef1a-mCherry-IRES-Cre) and then a viral vector driving a Cre dependent expression of an opsin (pAAV-Ef1a-DIO hCHR2(E123T/159C)-EYFP) was injected into the mPFC. This allowed conditional expression of the opsin only in neurons projecting to the DMS. We then implanted an optic fiber above the DMS to allow for the activation of mPFC terminals in the DMS and an optrode in the SC allowing the activation of mPFC terminals in the SC (of mPFC neurons projecting to

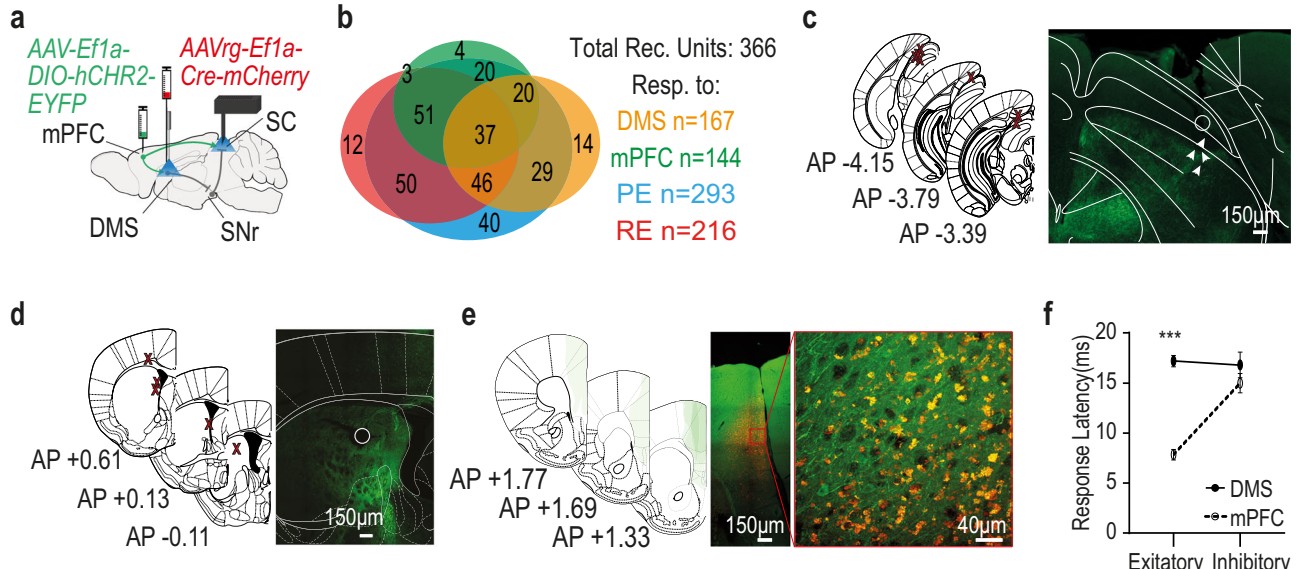

**Fig. 4 | SC neurons are affected directly or indirectly (through the DMS) by mPFC input. a** Schematic depiction of optrode placement (SC), optic fiber position (DMS), and viral injection sites (mPFC and DMS). **b** A Venn diagram representing the distribution of all recorded units in the experiment concerning DMS and mPFC responsiveness, as well as PE and RE responsiveness. Numbers on the diagram correspond to unit count. **c** Left: anatomical target schemes showing the SC. X's mark histologically confirmed most ventral position of the implant (either optic fiber or guide tube) from all 15 included animals (Representative images in this figure present results successfully acquired in all included animals) recording electrodes spread in a brush like form up to 1 mm below. Right: representative coronal section showing the SC with optrode lesion. White circle marks the suspected bottom of the optic fiber; white arrowheads indicate possible recording electrode location. **d** Left: histologically confirmed optic fiber placements at the DMS, marked by X's, from all included animals Right: representative coronal section showing the DMS with fiber lesion (White circle) and neural projections from the DMS) and recording SC neural activity (Fig. 4a). Out of the total 366 the mPFC in green fluorescence. **e** The left illustration depicts overlaid, histologically confirmed viral expression areas at the mPFC from all included animals. Right: representative coronal section showing viral expression and the co expression of red and green markers indicating DMS projecting neurons in the mPFC. **f** Latency of SC neural response to photo-tagging, divided according to the type of response (excitation/inhibition). The full line depicts phototagging of SC neurons responsive to activation of mPFC terminals in the DMS, dashed line are units responsive to direct activation of mPFC terminals in the SC. Error bars showing ±SEM. Only units excited by mPFC direct projection activation show a short latency typical to a direct synapse. Two-way ANOVA with Tukey's ($n = 144$ mPFC-positive, $n = 40$ mPFC-negative, $n = 129$ DMS-positive, $n = 37$ DMS-negative; mPFC-positive vs. DMS-positive–$p < 0.001$, mPFC-positive vs. DMS-negative–$p < 0.001$, mPFC-positive vs. mPFC-negative–$p < 0.001$). Asterisks indicate significant post-hoc comparisons (***$p < .001$). Source data are provided as a Source Data file. Parts of the Illustrations were generated with Biorender.com.

units recorded 166 responded to photo-tagging of the DMS, 144 were responsive to mPFC direct projections, both of those neuronal types were mostly reactive during PE (see Fig. 4b for the exact proportions of each category of response and Fig. 4c–e for expression and implanting maps in the different regions). While these are assumedly the same mPFC units which project both to the DMS and the SC we wanted to separate their direct and indirect influence on the SC neurons. Activating mPFC terminals in the DMS results in SC neural responses via the basal ganglia but might also result in antidromic activation which could lead to SC response via the direct input of the same neuron. Likewise, activating direct mPFC terminals in the SC could create an indirect SC response either via interneurons or antidromic activation. We therefore measured the response latency of SC neurons to light activation either above the DMS or directly above the mPFC and sorted according to the type of response (excitation/inhibition). We found a short latency corresponding to one direct synapse only for SC neurons excited by direct mPFC projection activation[43,44], while neurons inhibited by direct mPFC projection activation, and both types of responses occurring in SC neurons by activation of mPFC terminals in the DMS, showed a significantly longer latencies (Fig. 4f). A two-way ANOVA looking at the effect of response type (excitatory/inhibitory) and unit type (mPFC responsive/DMS responsive) on response latency after light stimulation found a significant interaction between the effects of response type and unit type was found ($F_{(1, 346)} = 20.32$, $p < 0.001$) with multiple comparisons tests revealing a significantly shorter response latency in excitatory mPFC-responsive units compared to inhibitory mPFC-responsive units and all DMS responsive

units (all $p < 0.001$). We therefore considered only short latency (<10 ms) SC neurons with excitatory responsive to direct SC light stimulation as direct mPFC responsive neurons and only SC neurons with long latency (>10 ms) response to light stimulation of mPFC terminals in the DMS, as DMS responsive neurons. Example responses of DMS responsive and mPFC responsive units could be seen in Fig. 5a and 5c respectively. Interestingly, when isolating the SC neurons responsive to those DMS projecting mPFC neurons, either through the direct mPFC-SC synapse or through the DMS, we found that those neurons changed their response both around PE onset and relative to the distance from the beetle. The temporal distributions of DMS-responsive units around escape onsets changed from before to after the shock phase, as more units initiated their response earlier relative to PE onset (pre-shock: 66%, post 7 days: 92% and post 21 days: 89% – Fig. 5b top), indicating a quantitative shift in involvement in PE initiation (A significant main effect to time in Kruskal-Wallis test ($H_{(2)} = 22.74$, $p < 0.001$) with post-hoc comparisons finding a significant difference of 7 and 21 days post shock from baseline both $p < 0.001$. Moreover, DMS-responsive SC units also changed the time of response initiation from before to after the shock phase in relation to the distance from the beetle. After the shock phase response was initiated at greater distance from the beetle, reflecting the shift in the safety distance which was evident in the mice behavior (Fig. 5b bottom). ANOVA revealed a significant effect of time ($F_{(2,104)} = 5.478$, $p < 0.01$) with post-hoc comparisons finding a significant difference of 7 and 21 days post shock from baseline; $p < 0.01$ and $p < 0.05$, respectively).

The same differences in the same direction were found in SC units responding to direct mPFC terminal activation of the same mPFC to

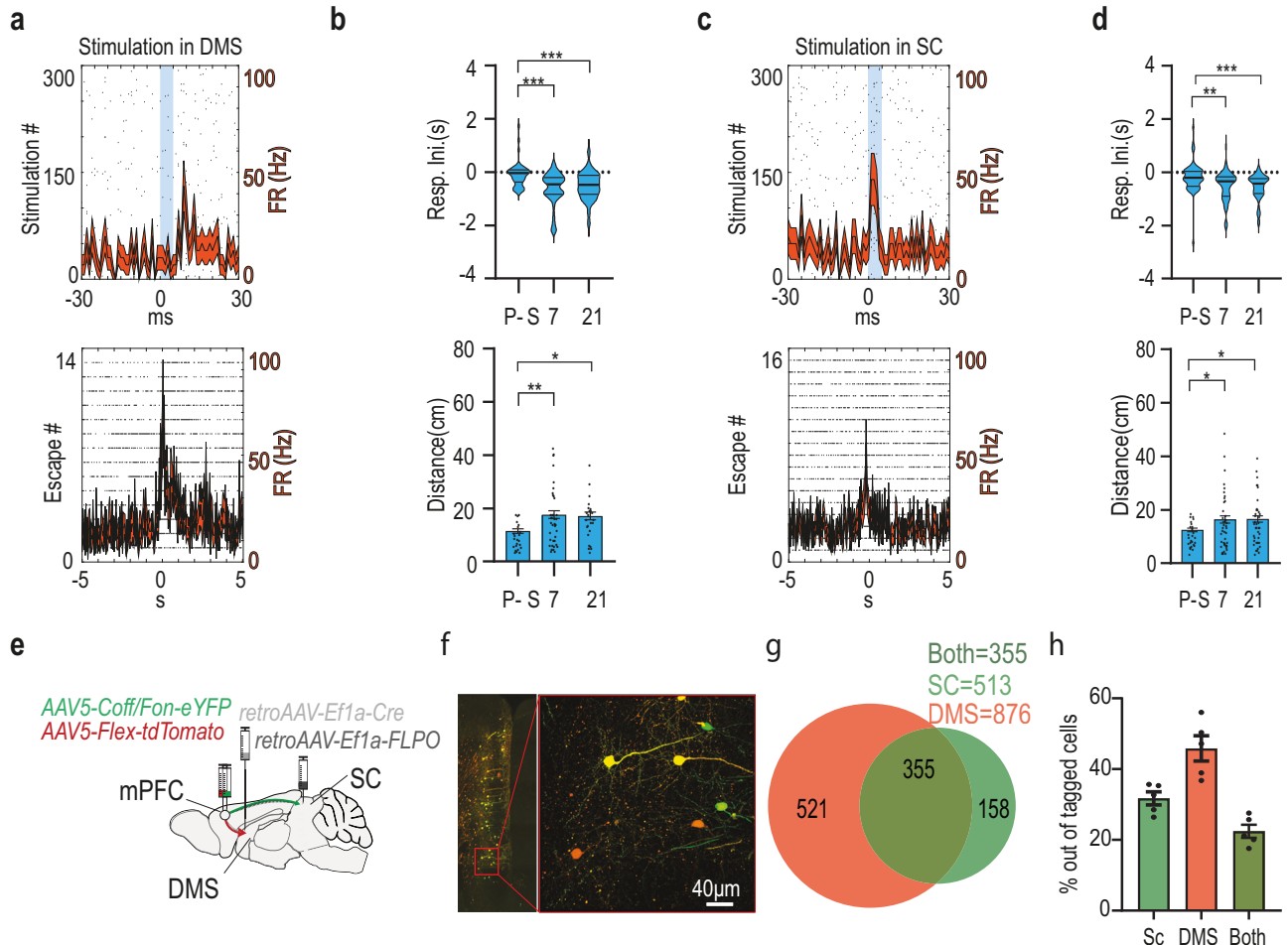

**Fig. 5 | SC neurons affected directly or indirectly (through the DMS) by mPFC input, change the neural response to support behavioral change after the shock phase. a** A raster plot overlaid with a PSTH showing a typical response of an SC neuron to light stimulation of mPFC terminals in the DMS (top) and to PE instances (bottom). **b** The units responsive to mPFC terminals activation within the DMS-Top: The distribution of response initiation times (ms) relative to PE onset in SC, undergoes shift afore following footshock. (Kruskal-Wallis with Dunn's (*n* = 44 pre-shock; *n* = 51 on 7th-day; *n* = 35 on the 21st-day; Pre-shock vs. 7th-day *p* < 0.001; Pre-shock vs. 21st-day *p* < 0.001). Bottom: An increase in distance (cm) from beetle corresponding to response initiation time following footshock. One-Way ANOVA with dunnet's (*n* = 29 pre-shock; *n* = 47 on 7th-day; *n* = 31 on the 21st-day, Pre-shock vs. 7th-day *p* = 0.004; Pre-shock vs. 21st-day *p* = 0.018). **c** Typical response of an SC neuron to light stimulation of mPFC terminals in the SC (top) and to PE instances (bottom). **d** The units responsive to mPFC terminals activation within the SC-Top: The distribution of response initiation times (ms) relative to PE onset in SC, undergoes shift afore following footshock. (Kruskal-Wallis with Dunn's (*n* = 58 pre-shock; *n* = 56 on 7th-day; *n* = 52 on the 21st-day; Pre-shock vs. 7th-day *p* = 0.002; Pre-shock vs. 21st-day *p* < 0.001). Bottom: An increase in distance (cm) from beetle

corresponding to response initiation time following footshock. One-Way ANOVA with dunnet's (*n* = 37 pre-shock; *n* = 52 on 7th-day; *n* = 50 on the 21st-day, Pre-shock vs. 7th-day *p* = 0.049; Pre-shock vs. 21st-day *p* = 0.039). **e** Top: A schematic depiction of viral injection sites: Retrograde AAV expressing Cre in the DMS and FLPO in the SC and the different recombinase dependent fluorophores at the mPFC. **f** Representative coronal section showing the mPFC with projection specific labeling in green (SC projecting), red (DMS projecting) and co-labeled neurons in yellow indicating that these neurons project to both (Successfully repeated in all 5 slices). **g** Venn-diagram representing the quantification mPFC neurons with projection specific labeling. **h** Bar plot representing the proportions of mPFC neurons with projection specific labeling out of all labeled neurons. In each plot, bars represent means of corresponding measures with error-bars showing ±SEM. In violin graphs, the bold midlines represent medians with IQR in regular lines above and below them. In PSTH, each line represents the overall average of the corresponding measure with shaded ±SEM. Asterisks indicate significant post-hoc comparisons (*\**p* < .05; \*\**p* < .01; \*\*\**p* < .001). Source data are provided as a Source Data file. Parts of the Illustrations were generated with Biorender.com.

DMS projecting neurons. We found a shift in the number of neurons initiating their response earlier relative to PE onset (Fig. 5d top). A Kruskal–Wallis test showed a significant main effect to time (H(2) = 25.44, *p* < 0.001), with post hoc comparisons revealing a significant difference at 7 and 21 days post shock from baseline (both *p* < 0.001). We also found that their neural response was initiated at a greater distance from the beetle after the shock, reflecting a shift in safety distance, which was evident in the mice behavior (Fig. 5d bottom). One-way ANOVA revealed a significant effect of time (F(2, 95) = 4.126, *p* < 0.05), with post hoc comparisons revealing significant differences at 7 and 21 days after shock from baseline (both *p* < 0.05). To confirm the presence of DMS-projecting mPFC neurons that also

branch to the SC, two different approaches were used. First, 3 animals were injected with green fluorescent microbeads (Retrobeads, Lumafluor) in the DMS and red retrobeads in the SC (Supplementary Fig. 2a, b); the other way around, 2 more animals were injected (Supplementary Fig. 2c, d). This allowed visual evidence of an abundant number of neurons in the mPFC, labeled with both red and green, suggesting that they were bifurcating mPFC neurons. However, because the microbeads aggregated within the cells, no reliable quantification of the number of those cells could be made using this method. Instead, retrograde viruses with different recombinases were used to specifically target projecting cell populations[44]. Three mice were injected with retrograde AAV expressing Cre recombinase into

the DMS and retrograde AAV expressing FLPO recombinase into the SC. Then, Cre-dependent red fluorophore (tdTomato) and FLPO-dependent green fluorophore (eYFP) were injected into the mPFC to conditionally express red fluorescence in DMS-projecting neurons and conditionally express green fluorescence in SC-projecting neurons. (Fig. 5e and Supplementary Fig. 2e). This approach allowed both visual detection (Fig. 5f) and quantification of the overlapping neurons. Out of the total 1034 tagged neurons, 158 were tagged in green (SC projecting), 521 were tagged in red (DMS projecting), and 355 neurons were tagged in yellow, indicating co-expression. This provides evidence for the bifurcation property (Fig. 5g), which maintained approximately the same proportions in all tested slices (Fig. 5h). Unlike for beads, which also tag projections, this method was also used to test for projection specificity in those neurons by measuring the fluorescence intensity in the injected and noninjected sides in all the measured slices of the DMS, SC and amygdala (Supplementary Fig. 2g–i). Indeed, compared to that in the noninjected side, there was significant red fluorescence in the injected DMS, green fluorescence in the SC and no fluorescence in the BLA. Three-way ANOVA of the changes in fluorescence intensity according to region (DMS/SC/amygdala), side (injected/noninjected) and fluorophore (EGFP/TdTomato) revealed a significant three-way interaction ($F(2, 60) = 8.251$, $p < 0.001$). Multiple comparisons tests revealed significantly greater fluorescence in the injected side than in the noninjected side at the DMS for TdTomato ($p < 0.001$) and a trend for EGFP ($p < 0.08$); similarly, for the SC, there was a greater expression of EGFP ($p < 0.01$), but no change was found in the amygdala (Supplementary Fig. 2f).

The change in the PE responses of mPFC and DMS responsive SC neurons was almost identical. Given that the mPFC neurons projecting to both regions are all identified as DMS projecting, it appears that these neurons have a unique contribution to the change in the safety distance maintained prior to escape. This idea is further supported by the results of testing the SC response to a general population of mPFC projection cells without restriction to DMS projection mPFC cells. Mice were injected with viral vectors driving the expression of light-sensitive proteins (AAV5-CamkIIa-hCHR2-EYFP) in projection neurons of the mPFC and implanted with an optrode at the ipsilateral SC (Supplementary Fig. 3a, d, e). Unlike in the experiment described before, the opsin expression in this case was not Cre-dependent. Thus, light stimulation above the SC evoked direct mPFC afferents, regardless of whether they also project to the DMS (or any other area) or not. Light-responsive units were termed 'non-specific mPFC-responsive' units (Supplementary Fig. 3b). We found that non-specific mPFC-responsive SC units have a dimmed effect both on the distribution of response and on the distance from the beetle at PE onset (Supplementary Fig. 3c left and right respectively), with a significant effect only 21 days after the shock, much like the general effect on all PE responsive SC neurons (Fig. 2). A significant main effect to time in Kruskal-Wallis test ($H(2) = 7.645$, $p < 0.05$) with post-hoc comparisons finding a significant difference only 21 days post shock from baseline both $p < 0.05$. One-way ANOVA revealed a significant effect of time on distance ($F(2,52) = 7.42$, $p < 0.01$,) with post-hoc comparisons finding a non-significant difference trend 7 days post shock ($p < 0.08$) and a significant difference 21 days post shock ($p < 0.001$) from baseline. This again indicates a distinct contribution of the branched pathway in the altered, post shock phase neural response to PE. On the post synaptic side, the physiological results (Supplementary Fig. 4) show that SC units responsive to both mPFC and DMS projection activation consistently differ in waveform (trough to peak latency) and in FR from SC units responsive only to mPFC projection activation, suggesting that the bifurcating mPFC target different SC cell types than other direct mPFC projection neurons. A one-way ANOVA comparing the trough to peak latency of the different responsive SC neuron groups found significant differences between the groups ($F(3,567) = 10.9258$, $p < 0.0001$) with post hoc tests finding mPFC responsive neurons to

differ from SNr responsive, DMS responsive and mPFC+DMS responsive SC neurons (all $p < 0.05$). A one-way ANOVA comparing the FR of the different responsive SC neuron groups also found significant differences between the groups ($F(3,567) = 6.0505$, $p < 0.001$) with post hoc tests finding mPFC+DMS responsive neurons again differ from mPFC responsive and SNr responsive neurons, and that DMS responsive and SNr responsive SC neurons also differed in their FR (all $p < 0.05$).

## Silencing of DMS projecting mPFC neurons during shock phase prevents the enhancement of defensive distance

The physiological findings indicated that DMS projecting mPFC neurons play an important role in controlling the enhancement of defensive distance kept by SC neurons involved in escape behavior. To test whether indeed those neurons are necessary to this process we used HM4D DREADDs (designer receptor exclusively activated by designer drugs) to selectively silence DMS projecting mPFC neurons during shock phase. Mice were injected with a retrograde viral vector driving the expression of the Cre bilaterally into the DMS (AAV-retro-hEF1a-iCre) and then a viral vector driving a Cre dependent expression of either HM4D for the test group (AAV-CaMKIIa-dlox-hM4D-mCherry) or a fluorophore for control (AAV-CaMKIIa-dlox-mCherry) bilaterally to the mPFC (See Fig. 6a for injection scheme and 6b for expression). The mice were put into the adapted BMT task and Clozapine-N-oxide (CNO) was administered to all mice (both groups) 20 minutes' prior the beginning of the shock phase in order to conditionally silence DMS projecting mPFC neurons during that phase (Fig. 6c). When examining the number of escapes, while we could not find any difference between mice expressing HM4D and Fluorophore controls in raw number of escapes (Fig. 6d left; Two way ANOVA interaction effect $F(2, 33) = 0.389$, $p = 0.68$, n.s.), when the number of escapes for each mouse both 7 and 21 days after shock was normalized to its own baseline (number of escapes prior to the shock), HM4D mice did not increase the number of their escapes as compared to controls (Two way ANOVA main effect for group $F(1, 33) = 4.389$, $p < 0.05$; Fig. 6d right). Likewise, we found that mice not expressing HM4D show the expected increase in defensive distance in PE 7 and 21 days following the aversive event. However, mice with selective silencing of DMS projecting mPFC neurons during the shock phase failed to show such an increase. Using a repeated measures mixed model ANOVA, we found a significant time X group interaction ($F(2, 35) = 3.268$, $p < 0.05$). Post-hoc test revealed a significant increase in PE defensive distance between pre-shock to 7 days ($p < 0.01$) and 21 days' post shock ($p < 0.05$) but not between any time point in the HM4D group (Fig. 6e). The same selective silencing of DMS projecting mPFC neurons during the shock phase prevented the reduction in the time spent in the center of the arena which control mice presented 7 days after the shock. Two-way mixed ANOVA effect of time $F(2, 34) = 5.05$, $p < 0.05$; multiple comparison found only P-S vs 7 days in control to be significant $p < 0.05$; Fig. 6f). This suggests that DMS projecting mPFC neurons are required during the adversity in order to develop such an aversive experience dependant increase specifically in defensive distance.

## Activating the SC terminals of DMS projecting mPFC neurons during the BMT controls defensive distance as a function of the distance from the approaching beetle

Finally, we tested whether those specific mPFC neurons which project to the DMS but also directly to the SC, are sufficient to change the defensive distance for escaping behavior, in naïve animals. Mice were injected with a retrograde viral vector driving the expression of the Cre bilaterally into the DMS (AAV-retro-hEF1a-iCre) and then a viral vector driving a Cre dependent expression of either an opsin for the test group (pAAV-Ef1a-DIO hChR2(E123T/T159C)-EYFP) or a fluorophore for control (pAAV-hSyn-EYFP) bilaterally to the mPFC. Optic fibers were implanted bilaterally above the SC. Thus, when stimulating, SC

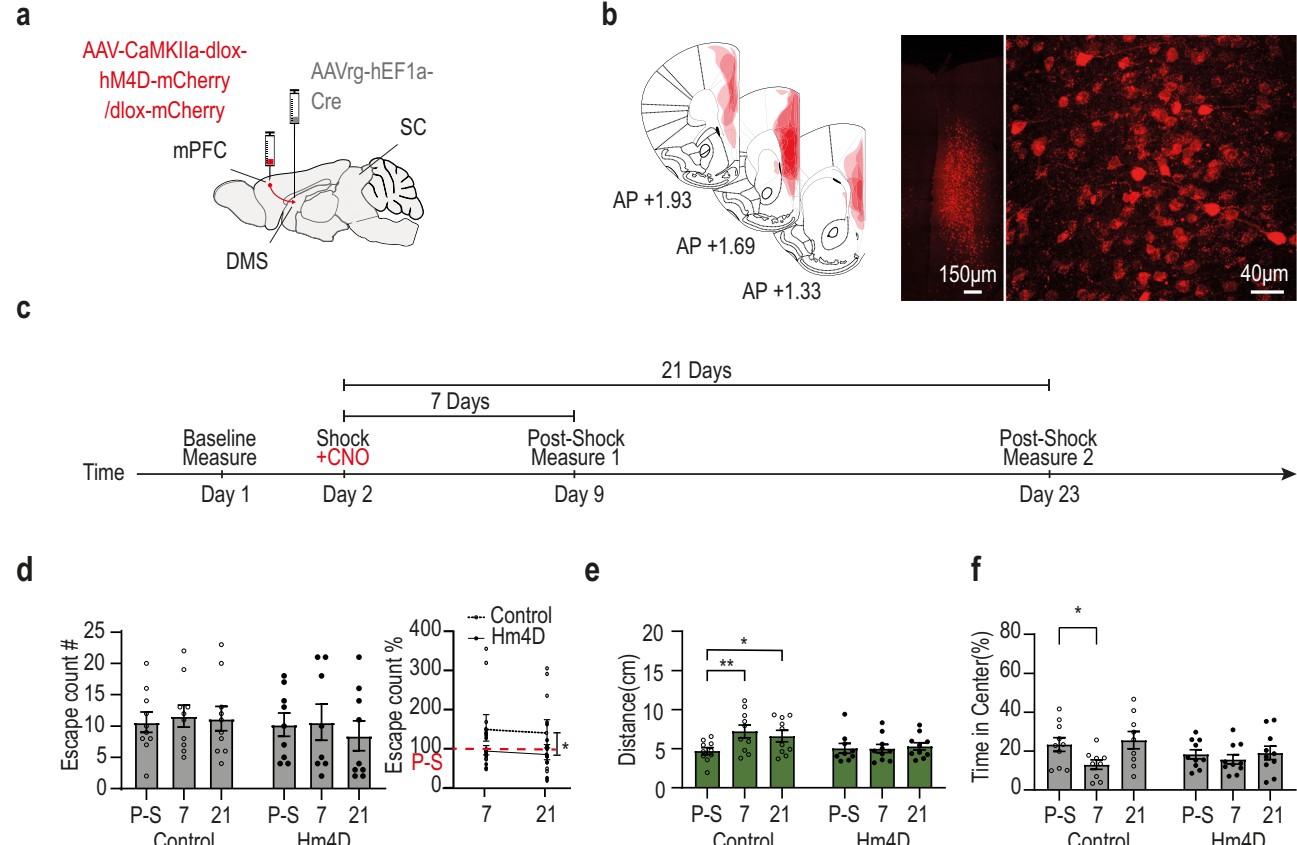

**Fig. 6 | Silencing of DMS projecting mPFC neurons during shock phase prevents the enhancement of defensive distance. a** Schematic depiction of control ($n = 10$) and HM4D ($n = 10$) viral injection sites (mPFC and DMS – representative images in this figure present results successfully acquired in all included animals). **b** The left illustration depicts overlaid, histologically confirmed viral expression areas at the mPFC from all included animals. Right: representative coronal section showing viral expression and neural cells in the mPFC. **c** Experimental schedule. All the following effects represent Two Way ANOVA with Tukey's post hoc comparisons. **d** Left: non-significant difference in the average number of escape counts animals showing long-term behavioral changes after the shock day, specifically in shocked mice ($p = 0.608$). Right: Testing the percentage change from baseline for each animal we found that HM4D mice increased the amount of escape instances

after shock as compared to control. Two Way ANOVA with main effect to HM4D ($p = 0.044$). **e** Increased defensive distance following shock phase in control animals, but no difference following shock phase in Hm4d group (Control: pre-shock vs. 7th-day $p = 0.002$; Pre-shock vs. 21st-day $p = 0.023$; HM4d: Control: pre-shock vs. 7th-day $p = 0.99$; Pre-shock vs. 21st-day $p = 0.92$). **f** A significant decrease in the time spent in the arena center was found 7 days after shock, only in control mice (Control: pre-shock vs. 7th-day $p = 0.043$; Pre-shock vs. 21st-day $p = 0.66$; HM4d: Control: pre-shock vs. 7th-day $p = 0.69$; Pre-shock vs. 21st-day $p = 0.97$). In each plot, bars represent means of corresponding measures with error bars showing ±SEM. Asterisks indicate significant post-hoc comparisons (*$p < 0.05$; **$p < 0.01$). Source data are provided as a Source Data file. Parts of the Illustrations were generated with Biorender.com.

afferents from the branched mPFC neurons are selectively activated (see Fig. 7a for injection scheme and Fig. 7c and d for fiber placement and expression in the SC and mPFC, respectively).

The behavioral measurement was divided into 4 counterbalanced phases (following a short habituation). Each phase lasted until 15 mouse-beetle encounters occurred upon which stimulation was activated to a total of 15 light stimulations (1 sec, 20 Hz; corresponding to the average duration and frequency of the response to escape recorded in the SC). In each phase the stimulation was activated at a different encounter distance: 10 cm, 20 cm, 60 cm (unrelated stimuli – this distance made sure that there was no real encounter during stimulation) and "none" (related to 15 beetle encounters without simulation – Fig. 7b).

To see whether the activation of the SC terminals of DMS projecting mPFC neurons is by itself sufficient to induce escape behavior, we compared the number of escape episodes (PE) of opsin and control animals occurring in each of the different phases. We indeed found that stimulation increased the number of escapes in the different phases, with the difference being focused on stimulation at 10 cm and at 20 cm phases (a repeated measures mixed model ANOVA found a significant main effect for opsin F(1, 13) = 6.468, $p < 0.05$; post-hoc

comparison showing opsin is different than control only at 10 cm and 20 cm phase, both $p < 0.05$). This indicates that the stimulation increased the number of escapes only when the beetle stimulus was drawing near. We could also see the near absence of escapes at the 60 cm phase, indicating that the mere activation of those projections without the presence of the beetle is not sufficient to initiate escape behavior (a significant main effect for test phase F(3, 39) = 22.09, $p < 0.001$; with Tukey's multiple comparison tests showing that animals at phase 60 cm had significantly fewer PE instances compared to all other phases; all $p < 0.001$; Also a main effect for group F(1, 13) = 6.468, $p < 0.05$; With post hoc tests showing the difference between control an d ChR2 is only at phases 10 cm and 20 cm, with both $p < 0.05$) (Fig. 7e).

As there were hardly any escapes at 60 cm phase, measuring the distance during escape onset was irrelevant. To assess the effect of activation of the SC terminals of DMS projecting mPFC neurons on defensive distance we analysed the distance between the mouse and the beetle on escape initiation during the three other phases. We found that in general the opsin expressing animals kept an increased defensive distance as compared to control fluorophore animals (a repeated measures ANOVA found a significant main effect for group

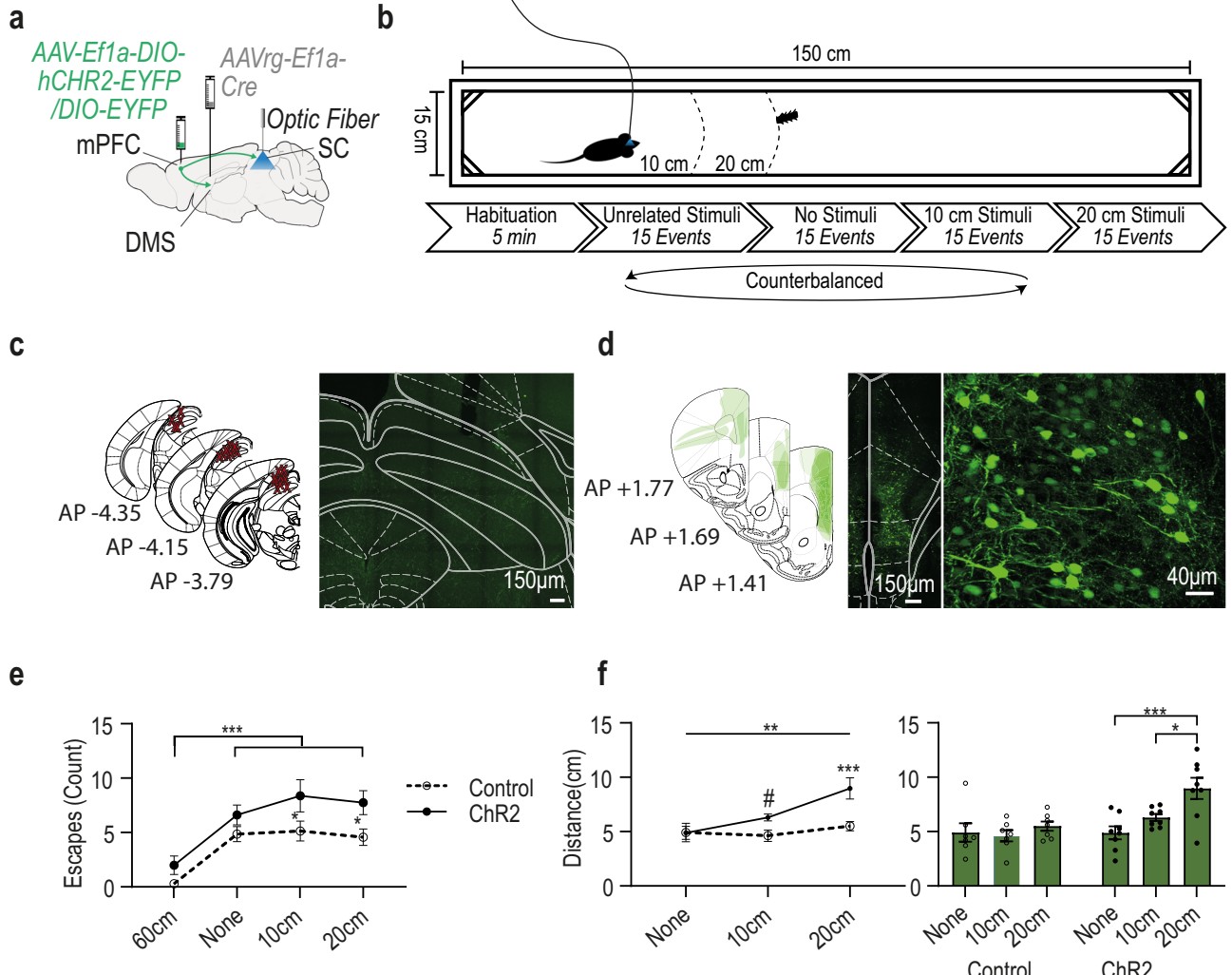

**Fig. 7 | Activation of the terminals in the SC of DMS projecting mPFC neurons during a beetle's approach affects defensive distance. a** Schematic depiction of control ($n = 7$) and CHR2 ($n = 8$) viral injection sites (mPFC and DMS) and fibers' location (Representative images in this figure present results successfully acquired in all included animals). **b** Experimental setup and schedule. **c** Left: anatomical target schemes for all animals included in the experiment. X's mark histologically confirmed optic fiber positions. Right: representative coronal section showing the SC with optic fiber lesion (white circle). **d** The left illustration depicts overlaid, histologically confirmed viral expression areas at the mPFC from all included animals. Right: representative coronal section showing viral expression and DMS projecting neurons in the mPFC. **e** Average (and SEM) of PE instances in each of the different stimulation phases divided to opsin expressing (ChR2) and fluorophore expressing (control) animals and showing an increase for ChR2 stimulation in the

different proximity distances. Two-way repeated measures ANOVA main effect with LSD's (Control none vs ChR2 none $p = 0.199$, Control 10 cm vs ChR2 10 cm-$p = 0.021$, Control 20 cm vs ChR2 20 cm-$p = 0.023$); **f** Average (and SEM) of the distance between the mouse and the beetle on PE initiation (defensive distance), in each of the different stimulation phases divided to opsin expressing (ChR2) and fluorophore expressing (control) animals shows increase in defensive distance for ChR2 group by stimulation distance. Two-way repeated measures ANOVA with Tukey's (ChR: none vs. 20 cm $p < 0.001$, 10 cm vs. 20 cm-$p = 0.025$; control vs ChR2: none-$p = 0.974$, 10 cm-$p = 0.08$, 20 cm-$p < 0.001$). Asterisks indicate significant post-hoc comparisons ($*p < .05$; $**p < .01$; $***p < .001$). Pound signs indicate non-significant trends ($^{\#}p < 0.1 - p > 0.05$). Source data are provided as a Source Data file. Source data are provided as a Source Data file. Parts of the Illustrations were generated with Biorender.com.

$F_{(1, 13)} = 13.70$, $p < 0.01$). The increase in defensive distance was a function of the distance on which stimulation was performed. That is, the larger the distance was between the mice and the beetle when stimulation commenced the larger was the defensive distance kept before escape was initiated (a significant main effect for test phase was also found ($F_{(2, 26)} = 6.07$, $p < 0.01$) with post-hoc comparisons showing that animals at phase 20 cm kept significantly increased defensive distances compared to phase 10 cm as well as to phase none ($p < 0.05$ and $p < 0.01$ respectively). Furthermore, a statistical trend in the interaction effect was also found $F_{(2, 26)} = 3.15$, $p = 0.06$ in which post-hoc comparisons revealed that only in the opsin condition animals at phase 20 cm kept significantly increased defensive distances compared to phase 10 cm as well as to phase none ($p < 0.05$ and $p < 0.001$ respectably), while no differences were found within the

control group. Additionally, it revealed increased defensive distances between opsin and control only at stimulation phases: at phase 20 cm ($p < 0.001$), as well as a statistical trend to increase at phase 10 cm between test control animals ($p = 0.07$; Fig. 7f).

These findings indicate that the DMS and SC branching mPFC neurons are sufficient to induce escape behavior in response to an approaching stimulus. Not only that, but they are also sufficient to set a defensive distance as a function of the timing of their activity.

## Discussion

The present study was designed to find the physiological basis of top down, long term changes in innate threat response, occurring after a significant aversive event, as reported in cases of PTSD[9,10,12]. Specifically, how does a traumatic event, such as a severe set of inescapable

foot-shocks, create long term changes in escape responses to perceived threat and the mechanism by which such changes are generated in the SC by mPFC influence.

Indeed, we found significant changes in innate escape behavior of mice long after the aftermath of the shock procedure. Not only did mice escape more frequently from a neutral, previously non-threatening stimuli (the robo-beelte) but they also escaped earlier, when the beetle was more distant, thus keeping an increased safety- or defensive distance compared to pre shock. This suggests an enhanced sensitivity to potential threats and a decreased threshold to initiate escape.

Two aspects of the changes in escape behavior, recorded in this study associate it with PTSD-like maladaptive changes. The first is the duration of the change – The DSM-V defines symptoms of PTSD that are consistent for at least 2 weeks and are ongoing at least 1 month after trauma exposure are considered to have a diagnosis of PTSD[45]. It has been argued that this period is much shorter for animals with a much shorter lifespan[46,47]. In that sense, the described changes in escape behavior were all long lasting and lasted for at least 21 days. A second aspect characterizing maladaptive fear behavior in PTSD is whether it is excessively generalized and inappropriately triggered[48]. The initially adaptive phenomena of "attentional enhancement due to potential threat" for example could also become maladaptive as exaggerated attentional bias towards threat-related information[49] could be directed towards more general unrelated information[50,51]. Interesting findings arising from the BMT indicate that the behavioral change after the aversive event is directed towards the robo-beetle which at no stage during the experiment posed a threat and during the baseline measurement prior to the shocks, was mostly not treated as one. Therefore, the prolonged changes in threat behavior aimed towards a non-threatening stimulus indicate a maladaptive nature.

While most effects were general changes to all escapes, the effects of adversity on defensive distance were specific to preventive escapes (PE) – corresponding to instances where visual detection of the incoming threat preceded the elicited escape (mouse facing the approaching beetle), and did not appear in reactive escapes (RE) – where the beetle approached the mouse from the rear (and the mouse did not turn its head toward the beetle), therefore not establishing visual contact prior to escape. This suggested that the change in defensive distance indeed requires visual detection.

Therefore, it might not be surprising that physiological changes supporting the shift in threat detection were indeed found in the SC, a midbrain structure known to integrate direct visual information and promote responses to visual threats[4,17–20]. Our findings support the involvement of this structure in innate escape as 80% of the units recorded in this area responded with increased neural firing rate to the animals' sudden acceleration onsets, while a vast majority thereof (87%) initiated response prior to acceleration onsets, suggesting SC involvement in the generation of this behavior. When looking at the exact initiation time of the neural responses of those neurons relative to escape onsets, we found that an increasing number of them shifted response initiation to an earlier time prior to escape initiation and to a further distance from the perceived threat long after the adversity. Changes in the physiology of the innate defensive system after a traumatic experience was already described both in animals and in humans[52,53]. Yet the systems driving such a change are still uncharted.

When assessing the contribution of different structures connecting into the SC, to the observed effects, we found that the quantitative change in neural engagement following shock, as well as the temporal and proximal shift of SC escape neurons could be clearly attributed to a unique set of neurons in the mPFC. The mPFC is thought to play a role in behavioral control related to emotions[27–29,54,55] and to induce behavioral changes due to changes in the emotional saliency and valence of a stimulus[56]. It also connects to the SC through its vast projections into the basal-ganglia[34,35] where it is thought to be the affective propagator

of goal-directed behavior[57,58]. Indeed, when testing the input of the mPFC into this system, by looking at the input reaching the SC from mPFC neurons projecting into the input nucleus of the basal ganglia – the DMS, we identified that SC escape neurons responsive to this input shift their response to support early escaping behavior. We also found that the effect of those DMS projecting mPFC neurons reached the SC both through the basal ganglia but also directly, and that though both ways, it supported the shift in the defensive distance. This effect was specific to those DMS projecting mPFC neurons, as when testing the general effect of SC neurons to direct mPFC projections, the effect was significantly milder. Also, when testing all input reaching the SC escape neurons from the basal ganglia through the SNr we could not find the specific shift after the shock treatment. The SNr is one of the main output regions of the basal-ganglia and the main gateway from the basal-ganglia to the SC and inhibition of this projection has been shown to directly participate in innate threat behavior[59] and induce active avoidance[35]. Our results also show that indeed the majority of recorded SNr responsive SC neurons are escape responsive, and that this escape related activity happens just prior to escape. The SNr is known to funnel and down sample information towards its output regions (including the SC) arriving from different regions and specifically from different basal-ganglia-thalamo-cortical circuits carrying different aspects of information[36,60,61]. The lack of shift in SC neurons responsive to general input from the SNr after the traumatic shock could indicate that this long term change is carried out by a specific cohort of SNr neurons rather than the entire SC projecting SNr neurons. Specifically, as our findings demonstrate, those neurons which receive information from the bifurcating mPFC neurons via the DMS. This dilution of the effect at the general population is supported by the dilution we see when considering all SC projecting mPFC neurons.

The specificity of the effect of DMS projecting mPFC neurons on the same escape neurons in the SC which shifted their response after the shock treatment, indicated that those unique mPFC neurons play a significant role in the physiological basis of the behavioral shift. The literature has long been describing long term changes in mPFC plasticity after fear[62], stress[63,64] and in relation to PTSD[65,66]. To test whether changes in the specific DMS projecting mPFC neurons contribute to the long-term change in defensive distance we selectively silenced them during the shock phase using DREADDS. We found that this selective silencing eliminated the long-term increase in defensive distance. This indicated that the activity of DMS projecting mPFC neurons during the adversity is necessary for this behavioral change to occur.

The indirect connection of those neurons to the SC through the DMS and the basal ganglia, could establish a long-term change in the selection of escape action. However, considering the importance of prompt and rapid responses to threat, an indirect connection to the SC could prove suboptimal for immediate threat response. Yet, through direct mPFC – SC connection by the same DMS projecting mPFC neurons, immediate influence can also be exerted. Notably, we found a bifurcating pathway, originating at the mPFC and branching both to the DMS and directly to the SC, confirmed both by our photo-tagging and electrophysiological recording, and by histological verification, using retrograde fluorescent tracing. We also found physiological evidence that these bifurcating pathways might be targeting a different set of SC neuron than direct, non-bifurcating mPFC projections.

Activating the direct terminals in the SC of the DMS projecting mPFC neurons of naïve animals running the task, at the average frequency and duration measured in the response of escape related SC neurons, resulted in an increase in the number of escapes but only when the beetle was actually approaching. This indicates that this pathway induces escape in relation to a target and not merely a burst in running velocity. Interestingly, the distance between the mouse and the robo-beetle in which the stimulation was activated affected the distance in which the escape was initiated at, indicating that the

activation of the direct SC projections of the DMS projecting mPFC neurons was sufficient to activate early escapes.

It is also worth mentioning that both silencing and activating the SC terminals of those unique mPFC neurons affected the defensive distance specifically and did not affect the velocity of the escapes (data not shown). This is interesting as downstream to the SC, a prominent tecto-nigral pathway affecting striatal dopamine[67] was shown to regulate locomotion speed in general and in association to a moving object specifically[68]. This tecto-nigral pathway was shown to increase the speed of approach behavior but not the latency of attacking[68] a moving prey. Interestingly, our results indicate a complementary mechanism which, while not affecting the velocity, specifically affects the latency to the escape.

Taken together the experiments outlined in this work show that changes occurring in the almost reflex-like escape innate defensive response, following a significant dire experience, map onto neurons in the SC. Specifically, SC neurons receiving input from a unique population of mPFC neurons that execute their effect on behavior both via the action-selection pathway of the basal-ganglia, but also directly on the SC. Those mPFC neurons can initiate escape from a perceived threat, and acquire alterations during adversity, rendering them more prone to earlier escape initiation through their connection to specialized neurons in the SC.

The mPFC in rodents and in humans, regulates adaptive fear behavior but plays a significant part in maladaptive behavior as well, especially in the context of PTSD symptoms[69]. Relevant symptoms to trauma-induced changes such as hyper-vigilance, hyper arousal and enhanced threat detection were all found to correlate with hyperactivity in what is termed the salience network whose cortical regions include an abundant portion of the anterior cingulate – the human homologues to the rodent's mPFC[70–72]. Increased activity in the dACC was found to be correlated with the tendency to avoid threat in PTSD patients[73], as well as persistence of fear in PTSD patients after extinction[74]. Interestingly, while the mPFC was indeed shown to be vital for adaptive behavior[54,75], aspects of this region seem to go off-balance after a traumatic event. Perhaps, as a region vital for adaptive behavior attempts to process an event which is too intense, it might undergo maladaptive changes, such as assigning excessive salience to a formerly neutral stimulus when guiding an action through the SC. The different functions, or purposes of the direct and indirect pathways from the mPFC to the SC, the dynamics of specifically projecting mPFC neurons after a traumatic event and whether other such pathways originating in the mPFC are serving different defensive actions, are questions for future research.

## Methods

### Animals
All experimental procedures were approved by the University of Haifa Ethics and Animal Care Committee, and the Israeli ministry of health (Authorization numbers 515/17 and 708/20) and adequate measures were taken to minimize pain and discomfort. A total of 90 C57BL/6 male Mice, (60 days old, -25 g; Envigo, Jerusalem, Israel), were housed together (up to 5 per cage) at $22 \pm 2\,°C$, 40-70% humidity, under 12-hour reversed light/dark cycles. Mice were allowed water and laboratory rodent chow ad libitum. For exact numbers per experiment see supplementary statistics table.

### Behavioral paradigm
**Beetle mania task (BMT).** Our modified version of BMT is based on a paradigm established at the Wotjak Lab it the Max Planck institute for Psychiatry[38]. First, mice are individually placed into a dimly lit, custom made, white acrylic arena (H37, L150 cm, W15 cm) (Fig. 1a, b) for 5 min habituation. Next, a randomly moving, battery operated toy beetle (H1.8 cm, L4.5 cm, W1.5 cm, weight: 7.3 g, mean speed: 25 cm/s; https://www.hexbeetle.com/nano, Innovation First Labs, Inc., TX, USA)

is introduced into the arena. For the next 10 minutes, the arbitrarily moving beetle creates accidental contacts with the mouse, allowing us to observe the animal's responses and general behavior. In our experiments this basic paradigm was first conducted with naïve animals to establish baseline measures and was repeated 7 days and 21 days following shock phase (described below) to examine possible differences in the behaviors of interest. Each trial was video-recorded in 25fps, using an overhead-mounted camera. In experiments involving electrophysiological recordings, 10 more minutes for photo-tagging were added to the schedule (5 after habituation and 5 at the end of the trial). In the final experiment the schedule was slightly altered (Fig. 6b for further discussion see the results section).

**Behavioral measures.** All measures regarding the animals' behavior were detected and extracted using DeepLabCut2[40] and a custom made code in Matlab (Version R2021a). The study focused primarily on aspects of escape behavior. An escape was defined as a sudden acceleration of the mouse, in reference to the approaching beetle (during all phases of acceleration). Later we made further distinction between escapes in which the mouse faces the approaching beetle, termed Preventive Escape (PE) and escapes in which the animal is approached from behind, termed Reactive Escape (RE). Where online detection of mouse-beetle proximity was needed, Ethovision XT (v14; 2018; Noldus) was used for this task. Statistical analyses were performed using Prism 8.0 (GraphPad) Statistica 13.05 (TIBCO) and Matlab (Version R2021a).Several behavioral measures were extracted:

**Number of escape occurrences.** Total count of escape instances as described.

**Defensive distance.** The distance between the mouse and beetle at escape onset. Our measure for proximal tolerance before threat response. The distance was measured slightly differently for PE as compared to RE because of the different angle of the mouse with the pose estimation markers and trying to minimize the difference stemming from the shape of the mouse. Thus, for PE the distance was measured from the beetle's center to the mouse head (to consider the distance from the visual sensing), whereas in RE as the beetle was arriving from behind, the distance was measured from the beetle's center to the mouse's center. This resulted in a slightly larger minimal distance in reactive escapes as compared to preventive escapes.

**Time in arena center.** Measure as the percentage of the total trial time in which the animal spent in the inner 40% of the arena (in accordance with the rectangular open field. This was used as an estimation for general anxiety.

**Stressor (shock phase).** To induce a non-associative, PTSD-like anxiety phenotype, we used the established dual shocks as a mouse model of PTSD[13,76,77]. Mice were individually placed into a shock chamber (Med-Associates, St Albans, VT), which was scented with 70% EtOH. After 198 s, two strong electric foot shocks (2 s, 1.5 mA) were presented, spaced apart by 40 s. Animals were returned to their home cage 60 s after the last shock. Control animals were placed into the shock chamber for 300 s, without shocks.

### Physiological procedures
**Stereotaxic surgery.** All mice were anesthetized by injection of a Ketamine-Xylazine Solution (0.04 ml/20 g, i.p.) and secured in a stereotaxic frame (Kopf Instruments). Isoflurane (1%) was delivered by a low-flow anesthesia delivery system (Kent Scientific) to maintain a deep anesthetized state over the course of the surgery. Body temperature was maintained at 37 °C by controlling their temperature with a thermal probe and an automatic controller system. Mice received an injection of analgesic agent Carpofen (0.01 ml/20 g, i.p.) and

ophthalmic ointment (Duratears) was applied on the mice eyes to protect the cornea from drying during the surgery. Following surgery, mice received systemic antibiotics (Amoxy LA, 0.04 ml/20 g, i.p.) for 4 consequent days (1 injection every 48 h) including the day of the surgery and were allowed to recover for at least 1 week before initiation of behavioral experiments. Depending on the specific experiment, optrodes, optic fibers, viral vectors or fluorescent tracers (Retrobeads, Lumafluor) were injected/implanted.

**Optrodes.** For electrophysiological recording and optogenetic stimulation at the SC, a custom made optrode was used, comprising of 16 tungsten wires and an optic fiber (Thor Labs), attached to an 18-pin connector (Mill-Max, NY). Optrodes were grounded via a wire placed in the cerebellum. The optrode was positioned unilaterally in the SC (AP:−3.80 mm, ML: ± 0.8 mm, DV:−1.5 mm) and secured to the skull using C&B Metabond (Parkell, Edgewood, NY) and dental cement.

Stimulation was conducted with a blue diode laser (CNI; $\lambda = 470$ nm), through an optical patch cord connected to the implanted fiber. To adjust for individual fiber light degradation, the light intensity for each optrode drive was measured with a calibrated power meter (ThorLabs) at the tip of the optical fiber before implanting. Light power was then measured before each experimental day at the tip of the optical patch cord at 100% intensity and adjusted to 16–20 mW at the tip of the implanted fiber.

**Recordings and stimulation.** Extracellular waveform signals were recorded using Digital Lynx hardware and software (Neuralynx Inc; USA). The signal was filtered (600-6,000 Hz) and amplified using a head-stage amplifier. Neural data was sorted manually using Offline Sorter (version 4.6.0; 2021; Plexon; USA) where a cutoff was applied to each original signal to leave only spikes deviating in size from the common signal decided upon by deviation from the histogram distribution of peak heights. Remaining spikes were sorted according to a combination of methods (including k-means, template matching and manual sorting) leading to separable clusters on a multidimensional space that includes principal components (2-3), peak to valley distance and stability over time of all dimensions considered. To look at characteristics of neural activity sorter output was further analyzed in Matlab (Version R2021a).

Response to light stimulation (Phototagging) was used to classify units responsive to specific inputs. 300 light stimulations ($\lambda = 470$, 5 ms wide pulses) were admitted through the optical fibers/optrodes at 0.5 Hz rate, 150 before and 150 after behavioral paradigm. Peri-stimulus-time-histograms (PSTH's) were created for each unit, with the time vectors stacked vertically and aligned to the stimuli at the matrix center column (creating an x-axis range of −30 ms to 30 ms around the stimuli, y-axis is the serial number of stimulus occurrence, patterning any neural firing around each stimulus). Paired-samples t-tests were calculated to compare the average baseline neural activity to the average activity in the following response time windows. For photo tagging the baseline window was based on the 30 ms prior to the delivery each light stimulation and was compared to 1 ms to 10 ms for an early response (omitting the first millisecond to prevent analyzing light on photoelectric artifacts[78,79]), and 11 ms to 20 ms for a late response. These ranges include the typical ranges reported for mono vs polysynaptic transmission following optogenetic axon stimulation[80–83]. As we have reported before, response latencies could be used in order to differentiate orthodromic from antidromic activation[55] and consider only the former.

For behavior – Units were classified as sensitive to certain events if their activity increased or decreased significantly around the event of interest. First, the neural spiking events were counted in 40 ms bins, to match the 25fps recording rate of the videos. Firing events vectors were then broken and arranged around the behavioral event of interest (e.g. start acceleration) from 5 seconds before each event to 5 seconds following the event to create a peri-event time histogram (PSTH) of the neural firing. To measure neural response to a behavioral event, we defined a baseline firing window in the PSTH lasting from 5 to 2 seconds prior to the behavioral event and compared it to the two seconds around the event (1 second before to one second after the event). Units with a significant comparison ($p < 0.05$) within that time were classified as responsive. Statistical analyses were performed using Prism 8.0 (GraphPad) Statistica 13.05 (TIBCO) and Matlab (Version R2021a).

**Viral vectors and beads.** Viral vectors were injected using a Nanofil Syringe mounted on a micro injection syringe pump and a (World Precision Instruments, Sarasota, FL). The injected volume was 600 nl at rate of 100 nl per minute. After each injection, the needle was left in place for an additional 1 minute per 100 nl and then slowly withdrawn. The skin was then resealed and mice were allowed to recover for at least 6-8 weeks to allow full expression of opsins. On unilateral injections vectors were injected ipsilaterally to optrode at the SC implanted later.

**List of vectors and beads.** For the investigation of the SNr-SC connection, pAAV-Syn-Chronos-GFP[84] (Provided by Edward Boyden, Addgene plasmid # 59170) was injected to the SNr (AP:−3.24 mm, ML: ± 1.4 mm, DV-4.5 mm), unilaterally, ipsilaterally to the implanted optrode at the SC.

For the investigation of the mPFC-SC connection, AAV5-CamKIIa-hChR2 (E123T/T159C)-eYFP[85] (Provided by Karl Deisseroth; UNC,NC) was injected to the mPFC (AP:1.70 mm, ML: ± 0.3 mm, DV-2.7 mm), unilaterally.

For the investigation of the mPFC-evoked DMS-SC connection, two viral vectors were injected. Into the mPFC (AP:1.70 mm, ML: ± 0.3 mm, DV-2.7 mm), pAAV-Ef1a-DIO-hCHR2(E123T/159 C)-EYFP[85] (provided by Karl Deisseroth, Addgene plasmid # 35509). The second vector, AAVrg-Ef1a-mCherry-IRES-Cre[86] (provided by Karl Deisseroth, Addgene plasmid # 55632) was injected into the DMS (AP: + 0.5 mm, ML: ± 1.25 mm, DV-3mm). Both vectors were injected unilaterally.

For chemogenetic inhibition of DMS projecting mPFC neurons, two viral vectors were injected. Into the mPFC (AP:1.70 mm, ML: ± 0.3 mm, DV-2.7 mm), a CRE-dependent vector, anterogradely driving the expression of hM4D receptors for Clozapine-N-oxide (CNO)-induced neuronal silencing, ssAAV5-mCaMKIIα-dlox-hM4D(Gi)_mCherry(rev)-dlox-WPRE-bGHp[87] (constructed by the VVF, Addgene #50461) or alternatively, for control animals, ssAAV5-mCaMKIIα-dlox-mCherry(rev)-dlox-WPRE-bGHp (constructed by the VVF) was injected. A second vector, retrogradely driving the expression of Cre, ssAAV-retro/2-hEF1α-iCre-WPRE-bGHp (constructed by the VVF, Addgene #24593) was injected into the DMS (AP: + 0.5 mm, ML: ± 1.25 mm, DV-3mm). Both vectors were injected bilaterally. CNO was administered (IP) to all mice at a dosage of 0.5/kg 30 minutes prior to the beginning of the behavioral measurement.

For optogenetic activation of DMS and SC projecting mPFC-afferents at the SC, two viral vectors were injected. Into the mPFC (AP:1.70 mm, ML: ± 0.3 mm, DV-2.7 mm), a CRE-dependent vector, anterogradely driving the expression of light-sensitive proteins, pAAV-Ef1a-DIO hCHR2(E123T/159 C)-EYFP[88] (provided by Karl Deisseroth, Addgene plasmid # 35509) or alternatively, for control animals, rAAV5/hSyn-Con/Foff--eYFP-WPRE[86] (provided by Karl Deisseroth, Addgene plasmid # 55651) was injected. A second vector, retrogradely driving the expression of Cre, ssAAV-retro/2-hEF1α-iCre-WPRE-bGHp (constructed by the VVF, Addgene #24593) was injected into the DMS (AP: + 0.5 mm, ML: ± 1.25 mm, DV-3mm). Both vectors were injected bilaterally.

For anatomical tracing of SC afferents and DMS afferents, Green and Red fluorescent tracer microspheres (Retrobeads, Lumafluor) were injected into the SC (AP:−3.80 mm, ML: ± 0.8 mm, DV:−1.5 mm) and into the DMS (AP: + 0.5 mm, ML: ± 1.25 mm, DV-3mm). The stock

solution was diluted in pbs (1/4) and the volume per injection area was 600 μL.

We also used a viral approach where four viral vectors were injected. Two recombinases were injected on retrograde viruses: retroAAV-Ef1a-Cre (provided by Dr. Karl Deisseroth, Addgene plasmid # 55636) to the DMS and Retro AAV-2-hEF1alfa-FLPo-WPRE-bGHp(A) (constructed by the VVF) to the SC. And two specific recombinase dependent fluorophores were expressed in the mPFC by: AAV5-hSvn-Coff/Fon-EYFP-WPRE (provided by Dr. Karl Deisseroth, UNC plasmid # AV6152B) and AAV5-CAG-Flex-tdTomato (provided by Dr. Ed Boyden, UNC plasmid # AV4599D). The recombinase dependent fluorophores were chosen to prevent leaking and cross-expression. While Coff/Fon is usually used for intersectional purposes on transgenic mouse lines, the use of retroviruses to carry the different recombinases could have lead to a slower expression of the recombinases and to different combinations of the recombinases in the protein pool at different time points eventually allowing co-expression as evident in the results.

### Reporting summary

Further information on research design is available in the Nature Portfolio Reporting Summary linked to this article.

## Data availability

The data generated in this study are provided in the Supplementary Information and Source Data file provided with this paper. All the data is available from the corresponding author upon reasonable request. Source data are provided with this paper.

## Code availability

Custom made codes used to analyze the data were made available on open science framework (OSF) at: https://osf.io/h7r4c/ and are available from the corresponding author upon reasonable request.

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

## Acknowledgements

We thank Carsten T. Wotjak for the discussion and ideas during the work on this project, Matthias Prigge and all the Klavir lab members for discussions and critical reading of the manuscript and Boris Shklyar from the Bioimaging Unit, Faculty of Natural Sciences, University of Haifa, for his help with the imaging. This work was supported in part by grants from the German Israeli Foundation (GIF I-270-421.10-2016; O.K), and the Israel Science Foundation (ISF 1495/17; O.K).

## Author contributions

O.K. directed the study. Experimental design, A.R and O.K, data acquisition, and interpretation, A.R, S.H., L.G., S.E and O.K.; manuscript writing, A.R. and O.K.

## Competing interests

The authors declare no competing interests.
