## [Peer Review File · Nature Communications]

Prefrontal control of superior colliculus modulates innate escape behavior following adversityREVIEWER COMMENTS

Reviewer #1 (Remarks to the Author):

This manuscript provides us an impressive perspective about neural circuit-level mechanism of PTSD. PTSD patients usually suffer from persistent psychosomatic disorder, one of which is characterized by alertness and susceptibility to startle, after catastrophic noxious trauma. In the mice model, the authors mimic these syndromes of PTSD patients with well-established escape paradigm. In view of the alarm feature, the author focuses on the SC, a neural structure which is organized into several laminae with convergence of inputs from multiple sensory modalities and elicits reliably aversive escaping behaviors, to dissect the potential mechanism of PTSD. On the other hand, some other substantial researches indicate that repetitive visual stimulation will result in adaptation of escape instead. In view of the moderation of visual threatening stimuli, the strong electric foot shock stimuli are considered to serve as a proxy for traumatic events. Effectively, the harmful memory after exposing shocks maintains for several weeks, even longer. Meaningfully, the authors sift a cluster of neurons in mPFC which bifurcate to both SC and DMS concurrently and prove the participation in mediating the long term avoidance-related memory to severe aversive events. The top-down effective modulation might greatly enrich our understanding of the neurobiology and neuropathophysiology in a more general sense. Some intriguing and feasible points might dig out in future work.

However, although the research theme is intriguing, there are still some issues I concerned yet to refer.

1. In figure 1, the foot-shock group performs a long-term change in escape behavior in day 7 and 21, whereas the non-shock control mice (figure 1h) do not reach a statistic significance. But the baseline of escape number (P-S) in non-shock mice is obvious lower than foot-shock group. It is reasonable to doubt that the no change of parameters (escape number, defensive distance, peak velocity) in control group is a result of the hyposensitivity to threatening stimulus.

2. In figure 1i and p, the authors divide the avoidance pattern into preventing and reactive escape. As figure 1r shows, the mice initiate escape at a defensive distance of more than 5 cm, neglecting of foot shock. However, what I am wondering is how the mice sense the approaching stimuli when the beetle is in blind zone of visual field and how the escape behavior is triggered when the robot beetle is behind the mice without contact.

3. In figure 2, the data indicates that a large fraction of SC neurons respond to acceleration in advance. But it is inevitable that the threatening visual stimuli of beetle dashing into mice is prior to acceleration in preventive escape. It is lack of correlation analysis of the preceding neural response with visual stimuli.

λ 4. As shown in figure 3e, after photostimulation of fiber terminals originating from SNr in SC, there is a certain degree of increase in SC neurons accompanying with decline subsequently. However, considering that the majority of neurons in SNr are GABAergic, I am confused about the increased firing rate of SC responding to inhibition effect from SNr.

λ 5. Compared the inputs from SNr in figure 3c with mPFC in figure 4c, it seems like that the two brain regions innervate SC in distinct manner. SNr project to the whole layers of SC (figure 3b) while the mPFC only to superficial layer (figure 4c). Likewise, the optrode placements in SC in the two brain regions distribute similarly. Thus, it need to be clarified that whether no efficiency of the SNr responsive SC neurons in PE is dimmed by the neurons in deep layers of SC. Of course, please preclude the possibility of fluorescence intensity firstly.

λ 6. In figure 4hj, the DMV responsive SC neurons perform a sharper significance than mPFC responsive neurons in PE. In consideration of possibility of antidromic and orthodromic activation through mPFC and SNr respectively, I am concerned that whether the SNr might make some contributions to the significance. If so, it may contradict the results about SNr in figure 3 above.

λ 7. A tectonigral pathway from SC to the substantia nigra pars compacta (SNc) is proved to enhance locomotion through dopamine release in dorsal striatum. In this study, it will be comprehensive to elaborate that whether the increased velocity after foot-shock stimuli is a consequence of the involvement of the cortico-striatal pathway or tectonigral-striatal pathway.

λ 8. What are the changes about response initiation latency of DMS-projecting mPFC responsive SC neurons to PE onset after chemogenetical inhibition in figure 5?

λ 9. There are many spelling and graphic mistakes in this manuscript, please correct it, figure 1dh, figure 2e, figure 4e legend and so on.

λ

Reviewer #2 (Remarks to the Author):

Ritter and colleagues examined the neural mechanism of the long-term change in innate escape behavior by testing the projection pathway from the medial prefrontal cortex (mPFC) to the superior colliculus (SC). They showed that the neural responses to preventive escape become earlier in the SC, in association with the change in escape behavior. They further identified that the bifurcating pathway to the SC from the mPFC that is also projecting to the dorsomedial striatum (DMS) significantly contributes to the behavioral change. While their findings are of interest, I request some clarifications from the authors by addressing the following issues.

Major issues:

1. The authors interpret the observed changes in escape behavior as PTSD-like maladaptive behavior. However, I was not convinced that the induced behavioral change was indeed maladaptive. One can claim that the increased sensitivity to a potential threat is rather related to attentional enhancement, which is adaptive and beneficial as it can facilitate flight from dangers. At least, the behavioral changes shown in Fig. 1 do not look as large enough as can be said as maladaptive.

This issue matters especially when it comes to the manipulation of PFC projection pathways. The authors showed that suppression of the PFC projection pathways prevents 'maladaptive' behavior, which seems odd because the PFC is in general thought to be vital for adaptive behavior. How can these inconsistencies be reconciled?

2. If the behavioral change is related to PTSD, the amygdala should also be relevant. Is it possible that DMS projecting mPFC neurons also project to the amygdala and that this projection mediates the behavioral change? This may affect their original conclusion.

3. Currently, only SC activity was recorded. Given that their main claim is related to prefrontal control of escape behavior, mPFC neurons should be examined. Do mPFC neurons also show defensive-related activity and do their activity properties change after shock exposure as seen in SC? Does the functional coupling between mPFC and SC also change after shock exposure?

4. Do the defensive-related activities in SC and mPFC have a contralateral bias as can be typically seen in their activity related to spatial attention?

5. How much is the overlap between mPFC neurons directly projecting to SC vs. those projecting to DMS? They only showed a representative histology section in Fig.4k, but there are no quantifications.

Minor issues:

1. The abstract is blurred and not informative, dominated by general background with very brief discrimination of results. It should include more specific results.

2. Information about some statistical procedures is lacking and should be more clearly described in the text. For example, what kind of "post-hoc test" is used after ANOVA; I saw lots of $p = \text{"ns"}$ and "No significant change" what are the p-values for " $p = \text{"ns"}$ " (Fig. 1h-k) and (Fig 3).

3. The number of animals used in each experiment should be described at least in Methods or the relevant Figure legends. Now readers have to guess from the histology panels.

4. The labels for x-axis are displaced in some figures (Fig. 1d,h, Fig. 2e).

5. The AP coordinates are not correct in Suppl Fig.1d.

Reviewer #3 (Remarks to the Author):

A Ritter et al., 2023 Nature Communications. Prefrontal control of innate escape behavior —A neural mechanism of enhanced posttraumatic threat response

Strengths:

In this manuscript by Ritter and colleagues, the authors investigate the long-range neuronal circuits that control threat detection and response to a motorized beetle stimulus in the prefrontal cortex (PFC), dorsomedial striatum (DMS), substantia nigra (SN), and superior colliculus (SC) following footshock using a combination of single unit recordings and opto/chemogenetics. Of particular interest are subpopulations of SC neurons that respond selectively to preventative escapes (PE) and reactive escapes (RE). The authors arrived at this conclusion from a series of phototagging experiments in the SC. Finally, the authors identify a projection from the PFC (containing DMS collaterals) to the SC that is necessary for the shock-induced enhancement of escape responses from the beetle. Although these results are of potential interest to the Nature Communications readership, there are a number of limitations in the current form of the manuscript that reduce enthusiasm. Below are suggestions to be addressed prior to publication.

Major concerns:

1) A primary claim in the title and discussion of this manuscript is the neural mechanism of enhanced posttraumatic threat response, but in the current version of the manuscript this claim is not supported. Although the authors provide several behavioral measures in Figure 1, the essential statistical comparison between shocked and non-shocked control mice is not reported. The authors observe time-dependent increases in the shocked groups in panels d-f. However, direct comparison of shocked and non-shocked groups for distance from beetle are not performed. The claims of stress-induced changes in threat detection hinge on observing differences between panels 1d-g and 1h-k. Moreover, the mean escape count in P-S between the two groups (1d and 1h) are quite different, raising concerns that prior to shock these two groups were already fundamentally different. Finally, the central claim of the manuscript is that prior shock experience heightens threat detection; yet visual comparison between the distance from beetle on day 21 between shocked and non-shocked groups (1e and 1i) shows they are quite similar. This raises concern that the increased distance from beetle reported in 1e is not necessarily related to the consequence of prior shock.

2) Regarding the separation of RE and PE, additional details in the methods would be helpful. From the methods it is not clear how exactly these escapes were identified. Visual detection by the mouse that the beetle is approaching (in the case of PE) would likely occur in a field of view larger than the 180 degrees shown in panel 1l (due to eye placement on the mouse). Additionally, in panels 1n and 1r, by visual inspection, it appears the distance from the beetle trends to be greater in RE compared to PE prior to shock. One would predict that a PE would occur at a greater distance compared to a RE. Additional details about how RE and PE would help clarify this concern.

3) The P-S to day 7 effect on distance from beetle for PE in figure 1n is not replicated in figure 2f.

4) Additional details in the methods regarding the single unit recording procedure are required. Currently, the procedure and criteria for spike sorting are not provided. Additionally, providing unit/spiking characteristics (i.e. mean firing rate, trough-to-peak latency) would be valuable. Regarding the phototagging procedure, it is not clear what exactly was compared to classify a cell as phototagged. Was it any spike within 1 to 10 ms after blue light delivery for early, and any spike within 10 to 20 ms for late? If this is the case, given the high basal firing rate shown in figures 3e, 4g, 4i, how did the authors control for false-positive photo-identification?

5) Figure 5 is a critical manipulation of the PFC-DMS-SC pathway and would be strengthened by showing all the dependent measures reported in Figure 1 (velocity, time spent in center, mean escapes). Also, were RE and PE separated for this experiment? Finally, it is not clear from the methods if control mice (not expressing the DREADD) were also injected with CNO or vehicle.

Minor concerns:

1) Better image stitching in figure 3b would help identify the electrode location. In other histology images that illustrate electrode/optic fiber location, adding white arrowheads to direct the reader to the location would be helpful.

2) The range of distance from beetle values varies throughout the manuscript, from approximately 5 to 10 cm in some figures (1 and 5) and around 15 to 20 cm in other figures (2, 3, 4). Behavioral measures in general can be variable and are expected to be so, but was there an alteration in the experiment that produced these differences?

3) No quantification is provided to demonstrate the specificity of labeling in panel 4k. Some quantification that demonstrates the strength of this overlap is required.

4) How exactly were acceleration-sensitive units defined? Under what circumstances was the pre-event baseline period 2 or 5 seconds? Also, were acceleration-sensitive units quantified in Figures 3 and 4?

5) Figure 6 legend does not contain statistical analysis information.

6) Light intensity used for optogenetic experiments is not reported. Dose of CNO and route of administration is not reported.

Detailed responses to reviewers

Dear Editors at Nature Communications,

We thank you and the reviewers for the constructive remarks. We have spent the last months conducting additional experiments, additional analyses, revising and improving the paper to address the reviewers' concerns. While the mission seemed very hard at times (and not because of scientific reasons), we are proud of the final product, in which many improvements were implemented thanks to the reviewers' valuable remarks.

Below please find detailed answers, *reviewers' comments in italics*, **our response in bold**. We marked with a yellow marker on the manuscript the changes that were made and stated in this letter (except the figures which were all changed but the marker interferes with the visual).

Reviewer #1:

We are thankful that the reviewer found the perspective brought in this manuscript to be impressive and intriguing and for the reviewer's highly important and constructive feedback which highly improved our manuscript.

1. *"In figure1, the foot-shock group performs a long-term change in escape behavior in day7 and 21, whereas the non-shock control mice (figure1h) do not reach a statistical significance. But the baseline of escape number (P-S) in non-shock mice is obviously lower than foot-shock group. It is reasonable to doubt that the no change of parameters (escape number, defensive distance, peak velocity) in control group is a result of the hyposensitivity to threatening stimulus."*

The reviewer is correct, indeed in the parameter of escape count it seems that the original pre-shock baseline group was lower in the control group, which could have been interpreted as a prior hyposensitivity by that group. As the behavioral result was persistent in all replications and as the escape count was comparable also where no shock was delivered in baseline of the optogenetic activation experiment (Figure6), we suspected that the reason for this specific difference in Figure 1 was the specific control group. We therefore repeated the behavioral experiment with a new control group of 13 mice. We now added the control group into the same ANOVA model, directly comparing the different time points between shock and no-shock groups. The adding of another variable into the analysis, reduced the power of the effect, yet we indeed found that in the number of escapes, the distance at escape initiation and time in the center, there are consistent changes due to the shock procedure which does not occur in the no-shock controls. The original velocity measure effect could not be recreated in the shock group this experiment, nor could we reproduce it in any of the other experiments – not the loss of function Chemogenetic experiment (Figure 5) or in the optogenetic activation experiment (Figure 6), we therefore decided to drop it completely. The results of the renewed behavioral experiment are described in Figure 1, the relevant results section and the discussion were also corrected accordingly.

2. *"In figure1i and p, the authors divide the avoidance pattern into preventing and reactive escape. As figure1r shows, the mice initiate escape at a defensive distance of more than 5 cm, neglecting foot shock."*

However, what I am wondering is how the mice sense the approaching stimuli when the beetle is in blind zone of visual field and how the escape behavior is triggered when the robot beetle is behind the mice without contact.”

We thank the reviewer for the requested clarification and will now detail the way those distances were measured. While for preventive escape the distance was measured from the beetle’s center to the mouse head (to consider the distance from the visual sensing), the distance in reactive escape was measured from the beetle’s center to the mouse’s center as there was no point in preferring any rear point over the other, and referring to the head could create a bias of head turning. This indeed resulted in a slightly larger minimal distance in reactive escapes as compared to preventive escapes where contact was measured at around 5 cm. We have now added a sector to the methods under the definition of defensive distance in the behavioral measures sector.

3. “In figure2, the data indicates that a large fraction of SC neurons respond to acceleration in advance. But it is inevitable that the threatening visual stimuli of beetle dashing into mice is prior to acceleration in preventive escape. It is lack of correlation analysis of the preceding neural response with visual stimuli..”

As the SC receives direct visual information the reviewer’s suggestion could be a good alternative explanation for the SC neural response before escape behavior. To test for this option, for each PE responsive neuron (which responds after a visually identified incoming beetle), we compared all neural responses to the incoming beetle that were followed by escape reaction to all responses to the incoming beetle that were not followed by escape. Supplementary figure 1a now shows a PSTH of the different normalized firing rate responses in all recorded SC PE neurons to the incoming beetle followed vs not followed by escape. A statistical comparison of the response shows that when it is followed by escape it is significantly higher than the no escape cases where almost no response was registered (Supplementary figure 1b). This analysis demonstrates that indeed those neurons respond only when escape behavior is involved and not exclusively to the stimuli of beetle dashing into the mice. We have now added this analysis to the results and the supplementary figures.

4. “As shown in figure3e, after photostimulation of fiber terminals originating from SNr in SC, there is a certain degree of increase in SC neurons accompanying with decline subsequently. However, considering that the majority of neurons in SNr are GABAergic, I am confused about the increased firing rate of SC responding to inhibition effect from SNr.”

We agree with the reviewer that the example in figure 3e could appear as an initial increase in firing rate however this was not indicated statistically as measured in the early response (see methods for response classification). As the reviewer indicates when finding an SNr responsive SC unit, the photo-tagging response was almost solely inhibitory and therefore only units responding with inhibition were considered as SNr responsive SC neurons (143 units out of 630 total recorded units in the SC). The appearance of immediate excitatory response could happen in some electrodes due to the known phenomena of photoelectric artifacts, caused by the light hitting the tip of the electrode^{1,2}. However, these artifacts should be easily disregarded by considering the early response as starting from 1ms post light activation, thus omitting the effect of the artifacts as they occur only with the immediate light on. We have now added this explanation of the early response being counted from one millisecond after

turning on the laser to our methods section under “Recordings”. We also replaced the example trace in figure 3e to a figure which does not include the suspected artifact.

5. *“Compared the inputs from SNr in figure3c with mPFC in figure4c, it seems like that the two brain regions innervate SC in distinct manner. SNr project to the whole layers of SC (figure3b) while the mPFC only to superficial layer (figure4c). Likewise, the optrode placements in SC in the two brain regions distribute similarly. Thus, it needs to be clarified that whether no efficiency of the SNr responsive SC neurons in PE is dimmed by the neurons in deep layers of SC. Of course, please preclude the possibility of fluorescence intensity first.”*

When looking at the original version of the representative coronal sections in figures 3b and 4c, at the size and resolution it was presented, the reviewer was right to think that the innervated layers are different. In both figures the axons were almost undetectable and in 4c it was completely overshadowed by a bleaching in the photo that seems like fluorescence on the top of the whole picture. We have taken new pictures under a confocal microscope of the slices and replaced both pictures to now show the actual fluorescence only in the axons. We find that both regions project to all layers of the SC. The optrode placement as well is misleading. The X’s marks the lower point of damage detected by the optrode. Usually it marks the bottom of the optic fiber which protrudes up to a mm from the guide tube which makes most of the damage. The electrodes’ damage itself is mostly undetected. The wires protrude up to a mm below the fiber tip in a brush-like pattern (see figure 2a and methods for optrode description) since blue light at the intensity used is effective in a sphere up to around 1 mm below the fiber tip³ so the recording sites and responsive units are spread through all SC layers in all experiments. We now added a clearer explanation to the histological figures and marked in the example slices the location of the fiber and electrodes when visible.

6. *“In figure 4hj, the DMV responsive SC neurons perform a sharper significance than mPFC responsive neurons in PE. In consideration of the possibility of antidromic and orthodromic activation through mPFC and SNr respectively, I am concerned that whether the SNr might make some contributions to the significance. If so, it may contradict the results about SNr in figure3 above.”*

We thank the reviewer for the requested clarification. We were as concerned with the issue of antidromic vs orthodromic activation as the reviewer. We therefore considered the latencies of responses (see Figure 4f) to make sure we are indeed looking specifically at orthodromic responses. This, in order to properly differentiate the direct mPFC influence on the SC from the indirect influence through the DMS. To this end we divided our photo-post-synaptic responses to ‘early’ (up to 10 ms from light stimulation) and ‘late’ (10-20ms from stimulation onset) responses. These ranges include the typical ranges reported for mono vs polysynaptic transmission following optogenetic axon stimulation⁴⁻⁷. As we have reported before, response latencies could be used in order to differentiate antidromic and orthodromic activation⁸. Antidromic activation of mPFC neurons by their DMS projections activation, should result in short latency excitatory response in the SC via direct mPFC synapse. Antidromic activation of mPFC neurons by their SC projections activation, or indirect inhibition (through activation of interneurons) should result in long latency response in the SC via indirect mPFC polysynaptic connection. It was a relatively small proportion of the responding units which did not answer the definition of orthodromic activation (Direct mPFC-SC projection stimulation: 34/144 [24%])

and 25/167 [15%] of the indirect mPFC-DMS stimulation responsive SC neurons), and we did not consider those responses in the presented effects. However, it is worth mentioning that the effects are preserved when including them in the responsive pool as well (We do not include those figures as they seem like an almost similar version of the results in Figure 4). A clarification for this issue was added in the methods under “Recordings”, and we are also dealing broadly with this issue at the results section describing figure 4f-j.

Regarding the contribution of the SNr - The SNr is one of the main output regions of the basal-ganglia and the main gateway from the basal-ganglia to the superior colliculus. Inhibition of this projection has been shown to directly participate in innate threat behavior⁹ and induce active avoidance¹⁰. Moreover, our data as well supports the role played by SNr projections to the SC in escape behavior by showing that the majority (64%) of recorded SNr responsive SC neurons are escape responsive (either PE or RE or both – Figure 3d). Also, that the response happens just prior to escape Figure 3f. The lack of a significant shift in the escape related activity when looking at SC neurons innervated by indistinct projections arriving from the SNr does not indicate that the relevant information does not pass through this hub, but that this perspective is too general to notice it. The SNr is known to funnel and down sample information towards its output regions (including the SC) arriving from different regions and specifically from different basal-ganglia-thalamo-cortical circuits carrying different aspects of information¹¹⁻¹³, contributing to the decision to carry out innate defensive behavior. According to our results part of that information arriving specifically from the DMS input, originating in the bifurcating mPFC neurons, carries the change in sensitivity after the traumatic experience. We believe that this is a very important discussion and was therefore added to the general discussion in the paper.

7. *“A tectonigral pathway from SC to the substantia nigra pars compacta (SNc) is proved to enhance locomotion through dopamine release in dorsal striatum. In this study, it will be comprehensive to elaborate that whether the increased velocity after foot-shock stimuli is a consequence of the involvement of the cortico-striatal pathway or tectonigral-striatal pathway.”*

Once again, we thank the reviewer for sparking an important discussion adding to the theoretical volume of the paper. We have indeed originally presented several behavioral consequences to the shock treatment; it all led us to the conclusion that innate escape behavior changes due to the traumatic event. One such measure was the velocity of escapes. As the reviewer rightfully underlines, there is a prominent tectonigral pathway affecting striatal dopamine¹⁴, which regulates locomotion speed in general and in association to a moving object (prey) specifically¹⁵. Interestingly this tectonigral pathway was shown to increase the speed of approach behavior but not the latency of attacking¹⁵. Our results show a mechanism that is somehow complementary to that mechanism on two levels. First, we could not replicate the effect of the traumatic event on the velocity of the escape in any of our experiments and therefore removed this result from the relevant figures. Chemogenetically silencing or optogenetically activating the mechanism also did not affect the velocity but did specifically affect the latency to escape. Second, while the tectonigral pathway was proved to promote appetitive approach locomotion, the mechanism described in our work promotes aversive avoiding behavior, namely escape. The interaction between those mechanisms is highly interesting both in the level of the SC which participates in such completing behaviors, and in the level of the striatal dopamine, which might

affect the DMS differently when promoting goal-directed vs innate more stimulus-response behavior. We now discuss this issue in the discussion.

8. *“What are the changes about response initiation latency of DMS-projecting mPFC responsive SC neurons to PE onset after chemogenetical inhibition in figure5?”*

The loss function experiment which includes chemogenetic inhibition was designed and conducted without any electrophysiological recordings as unlike optogenetics or recordings, chemogenetics allows measuring natural untethered behavior after a precise physiological manipulation, which we believe has higher added value, especially in due to the nature of escape behavior. We believe that the electrophysiological results together with the chemogenetic and optogenetic behavioral results establish the necessity and sufficiency of the connection between the SC and the DMS and SC projecting mPFC neurons for response initiation latency after the shock. While directly showing a change in the latency of onset of DMS projecting mPFC responsive SC neurons after trauma and the lack of that after inhibition, as suggested by the reviewer could somewhat strengthen the results, it is very hard to accomplish as it will require reliably following the same neurons for almost a month to compare the latency before and after inhibition/control (Chemogenetic inhibition occurs during the shock treatment while the behavioral measures were taken one to three weeks later) . We therefore chose the advantages of untethered testing with the feeling that the effects on those neurons’ response initiation latency are covered sufficiently by the other experiments.

9. *“There are many spelling and graphic mistakes in this manuscript, please correct it, figure1dh, figure2e, figure4e legend and so on.”*

We apologize for the mistakes. We went through the manuscript to find and correct all of them.

Reviewer #2:

We are thankful that the reviewer found interest in our findings and for the important discussion, adding theoretical depth to the manuscript.

1. *“The authors interpret the observed changes in escape behavior as PTSD-like maladaptive behavior. However, I was not convinced that the induced behavioral change was indeed maladaptive. One can claim that the increased sensitivity to a potential threat is rather related to attentional enhancement, which is adaptive and beneficial as it can facilitate flight from dangers. At least, the behavioral changes shown in Fig. 1 do not look as large enough as can be said as maladaptive.*

This issue matters especially when it comes to the manipulation of PFC projection pathways. The authors showed that suppression of the PFC projection pathways prevents ‘maladaptive’ behavior, which seems odd because the PFC is in general thought to be vital for adaptive behavior. How can these inconsistencies be reconciled?”

The reviewer’s claim here is divided into two issues, each sparking an interesting discussion that is incremental to the studies of psychopathology and the neural circuits underlying it. The first issue is

regarding the question of when an adaptive behavior stops being adaptive and ventures into the maladaptive realms. This is specifically relevant in the discussion of fear behavior as an adaptive behavior vs. fear pathologies such as PTSD. There is an ongoing discussion on that matter, yet some aspects are mostly agreed upon. The first is the duration of the posttraumatic symptoms - The DSM-V defines symptoms of PTSD that are consistent for at least 2 weeks and are ongoing at least 1 month after trauma exposure are considered to have a diagnosis of PTSD¹⁶. It has been argued that this period is much shorter for animals with a much shorter lifespan^{17,18}. All the induced behavioral changes were long lasting and lasted for at least 21 days, falling under the definition of the disorder. A second aspect that shifts “healthy” fear towards being maladaptive is whether it is excessively generalized and inappropriately triggered¹⁹. The attentional enhancement due to potential threat could also become maladaptive as exaggerated attentional bias towards threat-related information²⁰ could be directed towards more general unrelated information^{21,22}. One of the strengths of the results is that the behavioral change after the aversive event is directed towards the robo-beetle which at no point of the experiment posed a threat and during the baseline measurement prior to the shocks, was mostly not treated as one. We therefore believe that the prolonged change in behavior towards a non-threatening stimulus could be considered maladaptive. This important issue was added to the discussion chapter of the manuscript.

The Second issue regarding the role of the mPFC in adaptive and maladaptive behavior. This issue could also be divided into two separate but converging answers. The first is regarding the definition of the rodent’s mPFC, and the second is regarding the role of the mPFC in adaptive behavior and what happens in maladaptive behavior. The medial prefrontal cortex (mPFC) is a complex and heterogeneous region and while there is some debate regarding the homologies of the primates’ and rodents’ mPFC, it is mostly accepted that the rodents infralimbic (IL), prelimbic(PL) and cingulate cortex are in many aspects the homologues of Brodmann’s areas 32, 25 and the anterior parts of area 24 in the primate²³⁻²⁵. When considering those brain regions in rodents and in humans, they indeed seem to regulate fear behavior but also play a significant part in maladaptive behavior as well, especially in the context of PTSD symptoms²⁶ and specifically, symptoms relevant to the particular changes in escape behavior induced by the shock in our paradigm. Hyper-vigilance, hyper arousal and enhanced threat detection in PTSD were all related to hyperactivity in what is termed the salience network whose cortical regions include an abundant portion of those homologue regions in the anterior cingulate²⁷⁻²⁹. Increased activity in the dACC was found to be correlated with the tendency to avoid threat in PTSD patients³⁰, as well as persistence of fear in PTSD patients after extinction³¹. Interestingly, while the mPFC is indeed shown to be vital for adaptive behavior³², aspects of this region seem to go out of balance after a traumatic event. This could make a lot of sense, when thinking of the traumatic event as out of scope for the adapting ability of the mPFC. As a region vital for adaptive behavior attempts to process this extreme situation adaptation attempt could create maladaptive changes, for example assigning excessive salience to a formerly neutral stimulus when guiding an action.

We have now added the definition of the mPFC to the introduction and the important discussion regarding its role in PTSD symptomatology to the paper’s discussion.

2. *"If the behavioral change is related to PTSD, the amygdala should also be relevant. Is it possible that DMS projecting mPFC neurons also project to the amygdala and that this projection mediates the behavioral change? This may affect their original conclusion."*

This is an interesting point brought up by the reviewer, and we have conducted a full new experiment to answer this point together with point 5 made by the reviewer: "How much is the overlap between mPFC neurons directly projecting to SC vs. those projecting to DMS? They only showed a representative histology section in Fig.4k, but there are no quantifications."

In the original manuscript we indeed only showed the overlap in a representative coronal section showing the mPFC with co-labeled neurons with red and green retro-beads injected to the DMS and the SC (respectively). This indicated that these neurons project to both the DMS and the SC, but we indeed didn't quantify it. When coming to quantify the cell numbers we found that due to the way the beads aggregate within the cells we could not reliably decide on the borders of the cells and thus could not quantify the cell numbers. We repeated the beads experiment with three more mice, this time flipping the beads color (red in the SC and green in the DMS) and again could not reliably quantify the number of overlapping cells (See the new Supplementary figure 1 a-d for the retro-beads experiment). We presented the beads experiment in the figure to show the visual overlap and used another method for quantification. To do that we used retrograde viruses with different recombinases to target specifically projecting cells³³. That is, we injected three mice with a retrograde AAV expressing Cre recombinase to the DMS and retrograde AAV expressing FLPO recombinase to the SC. We then injected Cre dependent red fluorophore (tdTomato) and FLPO dependent green fluorophore (eYFP) to the mPFC conditionally expressing red fluorescence in DMS projecting neurons and conditionally expressing green in SC projecting neurons. This allowed us not only to see but also to quantify the overlap and the overlap proportions (See new figure 4 k-m). This again showed a significant overlap between DMS and SC projecting mPFC neurons. Unlike the beads which aggregate in the cell body, the conditional expression of fluorophore allows for whole cell expression including the axons and terminals. We were therefore able to check the question regarding co-projections in the amygdala. We compared the injected side to the non-injected in the DMS, the SC and the amygdala, and found both sides in the amygdala to be clean of either red or green projections. That is, we found no co-projections of mPFC neurons either DMS or SC projecting to the amygdala (See the new supplementary figure 1 e-i).

3. *"Currently, only SC activity was recorded. Given that their main claim is related to prefrontal control of escape behavior, mPFC neurons should be examined. Do mPFC neurons also show defensive-related activity and do their activity properties change after shock exposure as seen in SC? Does the functional coupling between mPFC and SC also change after shock exposure?"*

We agree with the reviewer that considering our current results, the dynamics of mPFC activity before and after shock exposure becomes an even more interesting question, especially mPFC neurons projecting to midbrain regions involved in innate defensive behavior. The scope of this work however is focused on the superior colliculus as a starting point as it is known to be involved in such defensive behavior and show for the first time that those innate escape behavior neurons could in fact be controlled by the mPFC, and the way this control changes after a traumatic experience at the level of

the neurons implementing the escape. The fact that our results intrigued the questions regarding the upstream mPFC and the way it exerts such control, proves the importance of that first step, as we are indeed conducting a whole new project to examine mPFC changes in activity dynamics due to traumatic experience, this will allow the examination of other downstream effects on specialized regions that will change not only defensive, but other specific behaviors as well.

4. *“Do the defensive-related activities in SC and mPFC have a contralateral bias as can be typically seen in their activity related to spatial attention?”*

This is yet another interesting point brought up by the reviewer. As the recording was done in freely moving and behaving mice, and as both the mouse and the beetle are constantly moving, the first challenge is to decide on a reference point (in time and space) to test for the contralateral bias. To try and address that, we examined the responses of all PE neurons recorded in the SC and referenced to location of the beetle relative to the mouse’s head at escape onset. Very interestingly we did find a significant effect to the lateralization, where responses to the beetle arriving from the contralateral side to the recording electrode were significantly stronger than when the beetle arrived from the ipsilateral side (Supplementary figure 1c-d). This nicely demonstrates that the side from which the threat arrives affect these SC escape neurons. This is even more interesting on the background of the analysis done to the request of reviewer #1 (point #3) where we found that the responses of PE neurons were not merely visual as the responses by PE neurons were exclusive to cases where escape followed the incoming beetle. Taking those two points together it seems that the lateralization of the visual input changes the neural response, but only if it is sufficient to induce escape. These two analyses now appear at supplementary figure 1 and are discussed in the results section and discussion.

5. *“How much is the overlap between mPFC neurons directly projecting to SC vs. those projecting to DMS? They only showed a representative histology section in Fig.4k, but there are no quantifications.”*

See our answer to point number 2 where we bundled the two points.

Minor Issues:

1. *“The abstract is blurred and not informative, dominated by general background with very brief discrimination of results. It should include more specific results.”*

We have now edited the abstract to shortly describe the details of the results.

2. *“Information about some statistical procedures is lacking and should be more clearly described in the text. For example, what kind of “post-hoc test” is used after ANOVA; I saw lots of $p = \text{“ns”}$ and “No significant change” what are the p -values for “ $p = \text{“ns”}$ ” (Fig. 1h-k) and (Fig 3).”*

3. *“The number of animals used in each experiment should be described at least in Methods or the relevant Figure legends. Now readers have to guess from the histology panels.”*

In order to improve transparency while keeping readability within the text, we added a comprehensive table, describing all used statistical tests and respective values, number of subjects, and descriptive statistics.

4. *"The labels for x-axis are displaced in some figures (Fig. 1d,h, Fig. 2e)."*

5. *"The AP coordinates are not correct in Suppl Fig.1d."*

We apologize for the mistakes. We went through the manuscript figures to find and correct all of them.

Reviewer #3:

We are thankful that the reviewer found our results to be of potential interest and for the reviewer's careful reading and important feedback which we feel helped in improving our manuscript.

1. *"A primary claim in the title and discussion of this manuscript is the neural mechanism of enhanced posttraumatic threat response, but in the current version of the manuscript this claim is not supported. Although the authors provide several behavioral measures in Figure 1, the essential statistical comparison between shocked and non-shocked control mice is not reported. The authors observe time-dependent increases in the shocked groups in panels d-f. However, direct comparison of shocked and non-shocked groups for distance from beetle are not performed. The claims of stress-induced changes in threat detection hinge on observing differences between panels 1d-g and 1h-k. Moreover, the mean escape count in P-S between the two groups (1d and 1h) are quite different, raising concerns that prior to shock these two groups were already fundamentally different. Finally, the central claim of the manuscript is that prior shock experience heightens threat detection; yet visual comparison between the distance from beetle on day 21 between shocked and non-shocked groups (1e and 1i) shows they are quite similar. This raises concern that the increased distance from beetle reported in 1e is not necessarily related to the consequence of prior shock."*

This concern was also raised by reviewer 1, please see our answer to this reviewer's first point. Shortly, we agree with the reviewer that indeed the original control group could have been interpreted as a different prior to the shock. As the behavioral result was persistent in all replications and as the escape count was comparable also where no shock was delivered in baseline of the optogenetic activation experiment (Figure6), we decided to repeat the behavioral experiment with a new control group of 13 mice and to add the control group into the same ANOVA model, directly comparing the different time points between shock and no-shock groups. Please see our full answer to reviewer1 point #1. The results of the renewed behavioral experiment are described in Figure 1, the relevant results section and the discussion were also corrected accordingly.

2. *"Regarding the separation of RE and PE, additional details in the methods would be helpful. From the methods it is not clear how exactly these escapes were identified. Visual detection by the mouse that the beetle is approaching (in the case of PE) would likely occur in a field of view larger than the 180 degrees shown in panel 1l (due to eye placement on the mouse). Additionally, in panels 1n and 1r, by visual*

inspection, it appears the distance from the beetle trends to be greater in RE compared to PE prior to shock. One would predict that a PE would occur at a greater distance compared to a RE. Additional detail about how RE and PE would help clarify this concern.”

We thank the reviewer for the chance to clarify since this issue was apparently unclear, as aspects of it were also raised by reviewer 1. Regarding the distances, please also see our answer to point 2 made by Reviewer 1. Shortly - While for preventive escape (PE) the distance was measured from the beetle’s center to the mouse head (to consider the distance from the visual sensing), in reactive escape (RE) the distance was measured from the beetle’s center to the mouse’s center (as a neutral center point). This indeed resulted in a slightly larger minimal distance in reactive escapes as compared to preventive escapes as full contact was measured at around 5 cm. We have now added a sector to the methods under the definition of defensive distance in the behavioral measures sector. Regarding the field of view, this is a very tricky question when both the beetle and the mouse are freely moving. Even in head-fixed or anesthetized animals with a controlled environment there are different ranges to the estimated field of view by each eye and how it maps onto cortex and SC ³⁴⁻³⁶. As in our current setup both the mouse and the beetle are freely moving, and we have no real way of verifying the exact location of the mouse’s ocular gaze especially in the resolution of our neural recordings (video recordings are much slower), we measured relying on the position of the mouse’s head towards the beetle. Therefore, the choice to rely only on behaviors within the range of 180 degrees in PE was the conservative choice, while it may cause the loss of data in some cases of detection in the far periphery of the visual field, it makes sure that most escapes considered as PE were indeed visually detected.

3. “The P-S to day 7 effect on distance from beetle for PE in figure 1n is not replicated in figure 2f.”

We agree with the reviewer that this issue needs clarifying. While figure 1n depicts a significant increase in the distance from beetle at the onset of the escape behavior, the picture in figure 2f describes the distance from beetle at the onset of the neural response of preventive escape (PE) related neurons in the SC. Compared to the behavior, the neural response which included all PE neurons in the SC showed a more subtle effect which was apparent only 21 days after shock, and started at a slightly further distance, which makes sense as we show that most of the neural responses by those neurons start prior to the actual escape. One of the interesting findings presented in our paper is that when we screen the pool of PE responsive SC neurons and look only at those affected directly or indirectly (through the DMS) by mPFC input, the change in the neural response sharpens to resemble the behavioral response. This implies that while a lot of SC neurons participate in escape behavior, those that are affected by the bifurcating mPFC neurons play a unique role in the change in defensive distance after trauma. We have now clarified the difference between the behavioral results and the physiological results in the figure legends and in the results sections describing figure 2.

4. “Additional details in the methods regarding the single unit recording procedure are required. Currently, the procedure and criteria for spike sorting are not provided. Additionally, providing unit/spiking characteristics (i.e. mean firing rate, trough-to-peak latency) would be valuable. Regarding the phototagging procedure, it is not clear what exactly was compared to classify a cell as phototagged. Was it any spike within 1 to 10 ms after blue light delivery for early, and any spike within 10 to 20 ms for late?”

If this is the case, given the high basal firing rate shown in figures 3e, 4g, 4i, how did the authors control for false-positive photo-identification?”

We now added details in the methods, both under Optrodes and under Recordings and stimulation, referring to the single unit recordings, sorting, and classification of units either by phototagging or by behavior. Thanks to the reviewer’s helpful remark, we also added a figure characterizing the recorded units on two dimensions – trough to peak latency and firing rate, and divide them according to their response to the different afferents (Supplementary figure 4. This yielded an interesting result now discussed in the results section and mentioned in the discussion. As for the phototagging, we did not consider any spike as a response but instead looked at the average baseline firing of the unit and compared it to the average firing during the defined potential response window, for each stimulation event, only statistically significant responding was considered as a responsive unit. This is now stated more clearly in the methods.

5. “Figure 5 is a critical manipulation of the PFC-DMS-SC pathway and would be strengthened by showing all the dependent measures reported in Figure 1 (velocity, time spent in center, mean escapes). Also, were RE and PE separated for this experiment? Finally, it is not clear from the methods if control mice (not expressing the DREADD) were also injected with CNO or vehicle.”

We now added the other dependent measures to figure 5 as requested, however, as the velocity measure did not replicate in the new version of the behavior, and in the control group of experiment 5, the velocity measure was removed from all experiments. Regarding the control of this experiment, all mice (HM4D and fluorophore control) were all injected with CNO as now clearly stated in both the results and the methods sections.

Minor Issues:

1. “Better image stitching in figure 3b would help identify the electrode location. In other histology images that illustrate electrode/optic fiber location, adding white arrowheads to direct the reader to the location would be helpful.”

We have now changed the microscopy picture in figure 3b for one with better stitching and better visibility of the SNr projections. As requested by the reviewer we also added white arrowheads to indicate possible placement of electrode fibers where electrode lesion is visible and a white circle to indicate optic fiber termination. We added the indicators and their meanings also to the relevant figure legends.

2. “The range of distance from beetle values varies throughout the manuscript, from approximately 5 to 10 cm in some figures (1 and 5) and around 15 to 20 cm in other figures (2, 3 , 4). Behavioral measures in general can be variable and are expected to be so, but was there an alteration in the experiment that produced these differences?”

This point made by the reviewer returns to our answer to major point 3 made by the reviewer. The reviewer rightfully identifies the difference in distance at the mouse escape onset (figures 1 and 5) from

the distance at neural escape response onset (figures 2, 3, and 4) which is bound to be a bit longer as we show that the neural escape response onset happens prior to the actual escape behavior. We made some slight changes in the text to make the distinction between the measures clearer. Please see also our response to major point 3.

3. *“No quantification is provided to demonstrate the specificity of labeling in panel 4k. Some quantification that demonstrates the strength of this overlap is required..”*

For the full answer please see our answer to point 2 and 5 of reviewer2. Shortly, while seeing the overlap using retro-beads we found that due to the way the beads aggregate within the cells we could not reliably decide on the borders of the cells and thus could not quantify the cell numbers. We repeated the beads experiment with three more mice, this time flipping the beads color (red in the SC and green in the DMS) and again could not reliably quantify the number of overlapping cells (See the new Supplementary figure 1 a-d for the retro-beads experiment). We used another method for quantification. To do that which is retrograde viruses with different recombinases to target specifically projecting cells³³ which allowed us not only to see but also to quantify the overlap and the overlap proportions (See new figure 4 k-m). This again showed a significant overlap between DMS and SC projecting mPFC neurons.

4. *“How exactly were acceleration-sensitive units defined? Under what circumstances was the pre-event baseline period 2 or 5 seconds? Also, were acceleration-sensitive units quantified in Figures 3 and 4?”*

Units were classified as sensitive to certain events if their activity increased or decreased significantly around the event of interest. First, the neural spiking events were counted in 40ms bins, to match the 25fps recording rate of the videos. Firing events vectors were then broken and arranged around the behavioral event of interest (e.g. start acceleration) from 5 seconds before each event to 5 seconds following the event to create a peri-event time histogram (PSTH) of the neural firing. To measure neural response to a behavioral event, we defined a baseline firing window in the PSTH lasting from 5 to 2 seconds prior to the behavioral event and compared it to the two seconds around the event (1 second before to one second after the event). Units with a significant comparison ($p < 0.05$) within that time were classified as responsive. This was true for the general response to acceleration as described in figure 2 c-e and in response to specific accelerations as in preventive escapes (PE) and reactive escapes (RE) from figure 2 f-g and onward. We have now elaborated on this description appearing in the methods section under – Recordings.

5. *“Figure 6 legend does not contain statistical analysis information.”*

We did not include the statistical analyses in any of the legends, instead, for the sake of readability we included the gist within the results section and following the revision and to improve transparency while keeping the readability, we now added a comprehensive table, describing all statistical tests and respective values used in all the figures.

6. "Light intensity used for optogenetic experiments is not reported. Dose of CNO and route of administration is not reported."

To measure individual fiber light degradation, the light intensity for each optrode drive was measured with a calibrated power meter (ThorLabs) at the tip of the optical fiber before implanting. Light power was then measured before each experimental day at the tip of the optical patch cord at 100% intensity and adjusted to 16–20 mW at the tip of the implanted fiber. Regarding the CNO dose for the HM4D experiment, CNO was administered (IP) at a dosage of 0.5/kg 30 minutes prior to the beginning of the behavioral measurement. We now added the stimulation information to the methods section under "Recordings and stimulation", and the CNO dose under "Viral vectors and beads".

Reference:

- 1 Cardin, J. A. Dissecting local circuits in vivo: integrated optogenetic and electrophysiology approaches for exploring inhibitory regulation of cortical activity. *Journal of Physiology-Paris* **106**, 104-111 (2012).
- 2 Cardin, J. A. *et al.* Targeted optogenetic stimulation and recording of neurons in vivo using cell-type-specific expression of Channelrhodopsin-2. *Nature protocols* **5**, 247-254 (2010).
- 3 Mahn, M., Klavir, O. & Yizhar, O. in *Handbook of Neurophotonics* 235-270 (CRC Press, 2020).
- 4 Petreanu, L., Huber, D., Sobczyk, A. & Svoboda, K. Channelrhodopsin-2–assisted circuit mapping of long-range callosal projections. *Nature neuroscience* **10**, 663-668 (2007).
- 5 Piñol, R. A., Bateman, R. & Mendelowitz, D. Optogenetic approaches to characterize the long-range synaptic pathways from the hypothalamus to brain stem autonomic nuclei. *Journal of neuroscience methods* **210**, 238-246 (2012).
- 6 Ren, J. *et al.* Habenula "cholinergic" neurons corelease glutamate and acetylcholine and activate postsynaptic neurons via distinct transmission modes. *Neuron* **69**, 445-452 (2011).
- 7 Xiong, Q., Oviedo, H. V., Trotman, L. C. & Zador, A. M. PTEN regulation of local and long-range connections in mouse auditory cortex. *Journal of Neuroscience* **32**, 1643-1652 (2012).
- 8 Klavir, O., Prigge, M., Sarel, A., Paz, R. & Yizhar, O. Manipulating fear associations via optogenetic modulation of amygdala inputs to prefrontal cortex. *Nature neuroscience* **20**, 836-844 (2017).
- 9 Almada, R. C. & Coimbra, N. C. Recruitment of striatonigral disinhibitory and nigrothalamic inhibitory GABAergic pathways during the organization of defensive behavior by mice in a dangerous environment with the venomous snake *Buthus alternatus* (Reptilia, Scorpiones). *Synapse* **69**, 299-313 (2015).
- 10 Hormigo, S., Vega-Flores, G. & Castro-Alamancos, M. A. Basal ganglia output controls active avoidance behavior. *Journal of Neuroscience* **36**, 10274-10284 (2016).
- 11 Alexander, G. E. & Crutcher, M. D. Functional architecture of basal ganglia circuits: neural substrates of parallel processing. *Trends in neurosciences* **13**, 266-271 (1990).
- 12 Alexander, G. E., DeLong, M. R. & Strick, P. L. Parallel organization of functionally segregated circuits linking basal ganglia and cortex. *Annual review of neuroscience* **9**, 357-381 (1986).
- 13 Kolomiets, B., Deniau, J., Glowinski, J. & Thierry, A. Basal ganglia and processing of cortical information: functional interactions between trans-striatal and trans-subthalamic circuits in the substantia nigra pars reticulata. *Neuroscience* **117**, 931-938 (2003).
- 14 Dommett, E. *et al.* How visual stimuli activate dopaminergic neurons at short latency. *Science* **307**, 1476-1479 (2005).

- 15 Huang, M. *et al.* The tectonigral pathway regulates appetitive locomotion in predatory hunting in mice. *Nature communications* **12**, 4409 (2021).
- 16 DSM, V. & Association, A. P. (American Psychiatric Publishing, 2013).
- 17 Rau, V. & Fanselow, M. S. Exposure to a stressor produces a long lasting enhancement of fear learning in rats: Original research report. *Stress* **12**, 125-133 (2009).
- 18 Siegmund, A. & Wotjak, C. T. A mouse model of posttraumatic stress disorder that distinguishes between conditioned and sensitised fear. *Journal of psychiatric research* **41**, 848-860 (2007).
- 19 Vahabzadeh, A., Gillespie, C. F. & Ressler, K. J. Fear-related anxiety disorders and post-traumatic stress disorder. *Neurobiology of Brain Disorders*, 612-620 (2015).
- 20 Fani, N. *et al.* Attention bias toward threat is associated with exaggerated fear expression and impaired extinction in PTSD. *Psychological medicine* **42**, 533-543 (2012).
- 21 Mathews, A. & MacLeod, C. Cognitive approaches to emotion and emotional disorders. *Annual review of psychology* **45**, 25-50 (1994).
- 22 White, L. K., Suway, J. G., Pine, D. S., Bar-Haim, Y. & Fox, N. A. Cascading effects: The influence of attention bias to threat on the interpretation of ambiguous information. *Behaviour research and therapy* **49**, 244-251 (2011).
- 23 Laubach, M., Amarante, L. M., Swanson, K. & White, S. R. What, if anything, is rodent prefrontal cortex? *eneuro* **5** (2018).
- 24 Vogt, B. A. & Paxinos, G. Cytoarchitecture of mouse and rat cingulate cortex with human homologies. *Brain Structure and Function* **219**, 185-192 (2014).
- 25 Heilbronner, S. R., Rodriguez-Romaguera, J., Quirk, G. J., Groenewegen, H. J. & Haber, S. N. Circuit-based corticostriatal homologies between rat and primate. *Biological psychiatry* **80**, 509-521 (2016).
- 26 Fenster, R. J., Lebois, L. A., Ressler, K. J. & Suh, J. Brain circuit dysfunction in post-traumatic stress disorder: from mouse to man. *Nature Reviews Neuroscience* **19**, 535-551 (2018).
- 27 Bryant, R. A. *et al.* Neural networks of information processing in posttraumatic stress disorder: a functional magnetic resonance imaging study. *Biological psychiatry* **58**, 111-118 (2005).
- 28 Lokshina, Y., Nickelsen, T. & Liberzon, I. Reward processing and circuit dysregulation in posttraumatic stress disorder. *Frontiers in psychiatry* **12**, 559401 (2021).
- 29 Seeley, W. W. *et al.* Dissociable intrinsic connectivity networks for salience processing and executive control. *Journal of Neuroscience* **27**, 2349-2356 (2007).
- 30 Fani, N. *et al.* Neural correlates of attention bias to threat in post-traumatic stress disorder. *Biological psychology* **90**, 134-142 (2012).
- 31 Milad, M. R. *et al.* Neurobiological basis of failure to recall extinction memory in posttraumatic stress disorder. *Biological psychiatry* **66**, 1075-1082 (2009).
- 32 Morey, R. *et al.* Fear learning circuitry is biased toward generalization of fear associations in posttraumatic stress disorder. *Translational psychiatry* **5**, e700-e700 (2015).
- 33 Fenno, L. E. *et al.* Targeting cells with single vectors using multiple-feature Boolean logic. *Nature methods* **11**, 763-772 (2014).
- 34 Li, Y.-t., Turan, Z. & Meister, M. Functional architecture of motion direction in the mouse superior colliculus. *Current Biology* **30**, 3304-3315. e3304 (2020).
- 35 Sterratt, D. C., Lyngholm, D., Willshaw, D. J. & Thompson, I. D. Standard anatomical and visual space for the mouse retina: computational reconstruction and transformation of flattened retinæ with the Retistruct package. *PLoS computational biology* **9**, e1002921 (2013).
- 36 van Alphen, B., Winkelman, B. H. & Frens, M. A. Three-dimensional optokinetic eye movements in the C57BL/6J mouse. *Investigative ophthalmology & visual science* **51**, 623-630 (2010).

REVIEWER COMMENTS

Reviewer #1 (Remarks to the Author):

This manuscript clarifies a top-down cortical modulation pathway encoding innate escape behavior. What spurred my interest is the potential neural circuit mechanism associated with PTSD. A specific sub-cluster of mPFC neurons bifurcating to DMS and SC is required for the trauma-related changes of innate escape behavior, suggesting a neutral target for developing clinical treatment. The fact that every behavior is a phenotypical manifestation of some well-orchestrated network activity has been validated and elaborated once again. The authors support their findings with some ingenious AAV strategies, the data are clearly summarized. But some figures are laid out with mistakes, for example figure 2e, as well as some spelling mistakes still exist, please check and correct. Considering underlying genetic vulnerability in PTSD risk, the incorporation of some research regarding genetic molecular mechanisms would augment future research, if deemed necessary.

Now I support the acceptance of this manuscript.

Reviewer #2 (Remarks to the Author):

I'm grateful that the authors took the time to address most of my concerns. The revised version is a significant improvement.

However, I was confused about their retrograde tracing experiments shown in Fig. 4k-n, and Supl. Fig. 2e-i. They injected retroAAV-Cre to the SC and retroAAV-FLPo to the DMS. Then AAV-Coff/Fon-EYFP and AAV-Flex-tdTomato was injected to the mPFC. I do not understand why the authors used Coff construct (AAV-Coff/Fon-EYFP). If I understand correctly, in this injection condition, eYFP and tdTomato should not be co-expressed in principle because tdTomato is Cre-dependent but eYFP is not expressed in the presence of Cre. However, there are so many co-labelled neurons (Fig. 4m. Both=355). This made me skeptical, and I found it challenging to fully trust these results and interpretations.

Minor points:

The observed activity difference in Supl fig. 1c,d appears subtle. While a t-test indicates significance, given that only the peak was analyzed, it will be worthwhile to consider a broader analysis of mean activity over a specific duration around the acceleration, unless there is a compelling rationale for focusing solely on the peak response.

Supl fig. 1. the color legend is opposite: (a) "escape (red) and when no no escape follows (blue)".

Fig. 6c, right. I cannot see "white circle" well in the panel.

Reviewer #3 (Remarks to the Author):

The description of experimental methods, figure legends, and analysis is greatly improved in this version of the manuscript (phototagging, chemogenetics, behavioral quantification). Moreover, the authors have ran additional behavioral experiments and AAV-mediated tracing studies. These changes have strengthened the clarity of the manuscript. Despite these positive changes, there are remaining concerns to be addressed.

1) The author's clarification about Fig 2f was helpful. However, if the claim is that the authors identified SC PE neurons, the prediction is that at 7 and 21 days, the activity of these SC should be

occurring prior to behavioral (PE) response initiation, compared to P-S. It is difficult to interpret the current results that only show significance at 21 days. Given the behavioral results in Fig 1i, the expectation is that PE neurons in the SC would reflect PE behavior in Fig 1i. If the neuronal activity does not mirror the distance from robo-beetle effect in Fig 1i, labeling these neurons as "PE neurons" may not be appropriate.

2) The additional details regarding the single unit recording procedure were helpful and the inclusion of Supp Fig 4 is valuable. However, there are still no criteria provided for the manual sorting of single units. Regarding the phototagging experiments, the authors provide several methods/protocol and empirical article references, but an explanation of how they controlled for false positives in phototagging is still absent. Given the high baseline firing rate they observed in SC units (as shown in Supp Fig 4), false-positives in phototagging is a legitimate concern that needs to be addressed, especially when the phototagging procedure uses 300 trials.

3) The authors state that the escape counts did not differ between groups in Fig 5d, so the escapes are calculated as a percent of baseline. However, manuscript consistently uses mean escape count (Fig 1d, h, k and Fig 6e). In the interest of comparability throughout the manuscript and as a demonstration of the internal replication of the inescapable shock and robo-beetle interaction, Fig 5d should be expressed as an escape count per animal, rather than a percent change from baseline. Additionally, given the circuitry investigated in Fig 5, and the results of Fig 4, the authors are likely only considering PE, however, it was not clear in the Results section that this was the case. Finally, Fig 5f control group comparison between P-S and 21 does not replicate the Time in Center (%) effect observed in Fig 1f.

Minor points

1) Fig 2e, left panel still contains a graphical error

2) The methods section for Optrodes states a 460 nm blue diode laser was used, but the Recording and Stimulation section states that a 470 nm laser was used for phototagging. It is unclear what is being used for phototagging experiments.

Detailed responses to reviewers

Dear Editors at Nature Communications,

Like the reviewers, we also feel the first round of review improved the manuscript significantly and we are thankful for the constructive remarks. As in the first round we take the remarks very seriously and implement the needed changes to the text and figures.

Below please find detailed answers, *reviewers' comments in italics*, **our response in bold**. We marked with a yellow marker on the manuscript the changes that were made and stated in this letter.

Reviewer #1:

We are thankful that the reviewer now supports the acceptance of the manuscript, much of the improvements were thanks to their constructive and helpful remarks.

1. *"Some figures are laid out with mistakes, for example figure 2e, as well as some spelling mistakes still exist, please check and correct."*

We apologize for the mistakes. We went through the manuscript figures to find and hopefully now correct all of them.

Reviewer #2:

We are thankful that the reviewer finds this version of the manuscript to be a significant improvement, many of the improvements were thanks to their thoughtful remarks.

1. *"I was confused about their retrograde tracing experiments shown in Fig. 4k-n, and Suppl. Fig. 2e-i. They injected retroAAV-Cre to the SC and retroAAV-FLPo to the DMS. Then AAV-Coff/Fon-EYFP and AAV-Flex-tdTomato was injected to the mPFC. I do not understand why the authors used Coff construct (AAV-Coff/Fon-EYFP). If I understand correctly, in this injection condition, eYFP and tdTomato should not be co-expressed in principle because tdTomato is Cre-dependent but eYFP is not expressed in the presence of Cre. However, there are so many co-labelled neurons (Fig. 4m. Both=355). This made me skeptical, and I found it challenging to fully trust these results and interpretations."*

The use of the AAV-Coff/Fon-EYFP was indeed unconventional. We carried the viral experiment in parallel to the replication of the retro-beads experiment, to add a complementary approach for testing the same issue. We chose to use two types of recombinases to get an expression in the mPFC that will specifically depend on the projection target (as comparable as possible to the beads) and while using FLEX for Cre dependent expression, in focusing too much on preventing leaking and cross-expression, we chose to use Coff/Fon construct for FLPo dependent expression. As the reviewer was concerned, this was done without enough consideration of the "off" switch. However, this does not change the fact that those constructs were used (Following the reviewer's remark we did some PCR sequencing to make sure of that), and that co-localization did happen. As also demonstrated in both figures, we do see green projections in the SC, red projections in the DMS, separated green and red cells in the mPFC and co-expressing cell in the mPFC as well. This cannot be explained unless those cells are projecting into both SC and DMS.

One possible explanation to the unconventional functioning of the intersect construct could be that although Cre “turns off” expression when conventionally using recombinase expressing transgenic mouse lines, here we used retroviruses carrying the different recombinase. Carrying by the retro viruses leads to a slower expression of the recombinases and to different combinations of the recombinases in the protein pool at different time points. This could allow co-expression, as although the eYFP will become restricted once the Cre expression stabilizes, the intersect construct “off” does not extinguish proteins that were already produced. This could explain both the co-expression and the numbers of co-labeled neurons.

Importantly, regardless of the functioning of this specific construct, the different colored projections in the SC and DMS with the co-labeled neurons in the mPFC undoubtedly demonstrate that those mPFC neurons project to both regions, as was the purpose of this experiment. This is on top of the fact that this viral approach was complimentary to the extensive evidence already accumulated in this work for the existence of those bifurcating mPFC neurons. Evidence stemming from microscopy of both the recording experiment and the beads experiments, as well as physiological evidence from the phototagging and electrophysiological experiment. We therefore hope that while unconventional, this method together with the rest of the experiments, did produce abundant evidence persuade the reviewer to trust our results.

However, in the case this is not satisfactory, we could repeat this experiment using a more conventional approach (same approach with regular fDIO instead of the Coff/Fon, which will take approximately 6-8 weeks to accomplish.

Minor Issues:

1.” *The observed activity difference in Supl fig.1c,d appears subtle. While a t-test indicates significance, given that only the peak was analyzed, it will be worthwhile to consider a broader analysis of mean activity over a specific duration around the acceleration, unless there is a compelling rationale for focusing solely on the peak response.*”

There are of course several ways to examine the neural response around acceleration. Since there are

variations in the timing of the response around acceleration onset (as described in figure 2), we thought the best way to capture the difference in response within each unit was not to look at a fixed common duration window but instead to consider the variance in the time window of response by finding and comparing the peak of each unit's response whenever it occurred (around the acceleration). As each response occurs at a slightly different timing around the event, the consideration of a fixed window should dim the response as it includes a considerable amount of time which is not an actual response. Indeed, conducting the same analysis but this time with the same time window, of one second around the acceleration initiation, for all units, we do get a weaker version of the same effect:

within subject ttest $t(840)=2.523$; $p<0.05$;

We therefore believe that the original version considering the peak is the more appropriate way to examine this specific data.

2. "Supl fig.1. the color legend is opposite: (a) "escape (red) and when no no escape follows (blue)"

Thank you, fixed.

3. "Fig. 6c, right. I cannot see "white circle" well in the panel."

Thank you, fixed.

Reviewer #3:

We are thankful that the reviewer found our manuscript greatly improved, many of the improvements were thanks to their insightful remarks.

1. "The author's clarification about Fig 2f was helpful. However, if the claim is that the authors identified SC PE neurons, the prediction is that at 7 and 21 days, the activity of these SC should be occurring prior to behavioral (PE) response initiation, compared to P-S. It is difficult to interpret the current results that only show significance at 21 days. Given the behavioral results in Fig 1i, the expectation is that PE neurons in the SC would reflect PE behavior in Fig 1i. If the neuronal activity does not mirror the distance from robo-beetle effect in Fig 1i, labeling these neurons as "PE neurons" may not be appropriate."

This issue touches on a fundamental and interesting issue in the results. where we see a behavioral phenomenon where predictive escapes (PE – escapes from a perceived incoming beetle) are launched at a further distance from the beetle after the traumatic event as the tolerated or safety distance

increases. We also identify neurons which change their firing rates consistently around such escapes. As their activity is specifically correlated with such escapes, we term them - PE neurons. Each individual behavioral escape response occurs when it occurs, closer or at further distance from the incoming beetle and yet the response of the PE neurons follows. Interestingly this response of firing rate change happens mostly before the escape response (as shown in figure 2). Another related phenomenon that we see, which the reviewer here relates to, is that this PE related neural response itself seems to shift in relation to the escape initiation. To us it was surprising – not only does the neural response happen at further distance from the beetle (as it is correlated with the escape) but it moves in relation to the related escape behavior. Indeed, when looking at all recorded SC PE neurons this phenomenon becomes significant only 21 days following the shock (which is already interesting), but when looking only at SC PE neurons responsive to the bifurcating mPFC neurons the phenomenon becomes distilled as this cohort of PE neurons initiate the neural response earlier than the behavioral response. Therefore, we think that while those neurons are now more sensitive PE neurons, they still maintain their PE related response, and are still PE neurons.

2. *“The additional details regarding the single unit recording procedure were helpful and the inclusion of Supp Fig 4 is valuable. However, there are still no criteria provided for the manual sorting of single units. Regarding the phototagging experiments, the authors provide several methods/protocol and empirical article references, but an explanation of how they controlled for false positives in phototagging is still absent. Given the high baseline firing rate they observed in SC units (as shown in Supp Fig 4), false-positives in phototagging is a legitimate concern that needs to be addressed, especially when the phototagging procedure uses 300 trials.”*

Regarding the sorting method, as requested by the reviewer we will try to further elaborate on the process: Recorded signal was uploaded into the plexon offline sorter, where a cutoff was applied to spikes deviating in size from the common signal expressed in deviation from the histogram distribution of peak heights. Remaining spikes were sorted according to a combination of methods (including k-means, template matching and manual sorting) leading to separable clusters on a multidimensional space that includes principal components (2-3), peak to valley distance and stability over time of all dimensions considered. This elaboration was now added to the methods under recordings and stimulation.

As for controlling false positives, after the first round of reviews we now describe the photo-tagging process in detail including all stages and the prevention of light artifacts. If the reviewer refers to statistical false positives, going through the literature we failed to find a standard method of controlling statistical false positives in photo-tagged extracellular neural responses, this issue is rarely discussed. We did of course go through all the PSTHs of our responsive units to make sure we visually believe it, but no statistical or mathematical method was used for statistical false-positive control. We do think that any false-positive units added to the data-pool will only burden our results by adding noise, and therefore we think that if anything, if there were statistical false positives, considered in our analysis this only strengthens our results.

3. *“The authors state that the escape counts did not differ between groups in Fig 5d, so the escapes are calculated as a percent of baseline. However, manuscript consistently uses mean escape count (Fig 1d, h,*

k and Fig 6e). In the interest of comparability throughout the manuscript and as a demonstration of the internal replication of the inescapable shock and robo-beetle interaction, Fig 5d should be expressed as an escape count per animal, rather than a percent change from baseline. Additionally, given the circuitry investigated in Fig 5, and the results of Fig 4, the authors are likely only considering PE, however, it was not clear in the Results section that this was the case. Finally, Fig 5f control group comparison between P-S and 21 does not replicate the Time in Center (%) effect observed in Fig 1f.”

Regarding mean escape count vs percent change from baseline. The reviewer rightfully states that we show the mean escape count throughout our manuscript. In figure 5 we chose not to show the mean escape count as the result does not replicate when considering the means as we clearly and explicitly state in the manuscript. There is an increase, but it is not significant. We do however find that the

result is replicated when looking at the change from baseline, so the phenomena where animals increase the number of escapes due to the inescapable shock persists, as it persists also in the experiments described in figure 1 and 6 as the reviewer states. We therefore have no doubt that the figure with the percent change should appear as it describes a real phenomenon, and it is the only thing that describes the persistence of the phenomena in the experiment described in figure 5. We have now added the statistics of this non-significant result in the results section where we state it is not significant and in the statistics table. We also added the bar-plot describing the non-significant mean escape followed by the figure of the percent change as shown in the figure added here.

Regarding which escapes are considered – We assume the reviewer refers to the results described in figure 5, and we therefore did some editing in the text to make it clearer.

Regarding the time in the center in figure 5 – Time in the center was considered as a general measure for anxiety much like the classical open field and differs from the innate escape behavior which we show to be affected by the mechanism identified in this paper. Already in figure 1f one might notice a slight increase on the 21st day as compared to 7 days post shock although still significantly lower than the baseline time in the center, which might indicate that the general anxiety on the 21st day is not as high as on the 7th day post shock. In figure 5f, with a significantly lower number of animals we do see the decrease in time spent in the center on the 7th day in control animals which does replicate the result obtained in figure 1, while we do not see this change in hM4D animals. However, it does seem that the level of anxiety returns to baseline on the 21st day in this cohort of control animals. This indeed differs from figure 1, this could happen due to the smaller number of animals, the different batch and other reasons. However, the effect of the shock does show on the 7th day, the direction is maintained, and it is negated by the treatment. This is an important result and what makes it even more interesting is that it shows that even though the effects of the shock on the general anxiety 21 days after the shock, are weaker (figure 1) or gone (figure 5), the effects on the innate threat behavior remain.

Minor Issues:

1. *"Fig 2e, left panel still contains a graphical error."*

Thank you, fixed.

2. *"The methods section for Optrodes states a 460 nm blue diode laser was used, but the Recording and Stimulation section states that a 470 nm laser was used for phototagging. It is unclear what is being used for phototagging experiments."*

We thank the reviewer. In this experiment we used the 470 nm laser. We fixed the Optrode section accordingly.

REVIEWERS' COMMENTS

Reviewer #2 (Remarks to the Author):

I appreciate the authors for taking the time to address my remaining concerns. With the information provided, I now feel that my concerns have been adequately clarified. I am satisfied, provided that the authors incorporate explanatory notes regarding their unconventional use of the AAV-Coff/Fon-EYFP construct, originally designed for intersectional purposes, and elucidating the reasons behind the observed co-labeling. This addition is crucial to preventing confusion among readers or potential recurrence of similar concerns.

Reviewer #3 (Remarks to the Author):

The authors have provided adequate responses to my concerns. Their explanation of PE neuron activity, their specific pathways, and the activity profile of this neuronal population during days 7 and 21 post-shock was very helpful.

The addition of single unit sorting methods will be appreciated by the readership. Regarding the control for falsely phototagged neurons. The authors rightly state that any falsely identified neurons added to the data pool would only burden their results.

Finally, the transparency the authors provide for their dependent measure in Figure 5 addresses my concerns.

Detailed responses to reviewers

Dear Editors at Nature Communications,

Below please find detailed answers, *reviewers' comments in italics*, **our response in bold**.

Reviewer #2:

We are thankful that the reviewer now supports the acceptance of the manuscript.

1. *"I am satisfied, provided that the authors incorporate explanatory notes regarding their unconventional use of the AAV-Coff/Fon-EYFP construct, originally designed for intersectional purposes, and elucidating the reasons behind the observed co-labeling. This addition is crucial to preventing confusion among readers or potential recurrence of similar concerns."*

We added the explanatory notes to the methods dealing with the specific viral approach.

Reviewer #3:

We are thankful that the reviewer finds the second revision helpful and now supports the acceptance of the manuscript.